# The RNA helicase DDX6 controls early mouse embryogenesis by repressing aberrant inhibition of BMP signaling through miRNA-mediated gene silencing

Jessica Kim[1], Masafumi Muraoka[2], Hajime Okada[2], Atsushi Toyoda[3], Rieko Ajima[2,4]*, Yumiko Saga[1,2,4]*

1 Department of Biological Sciences, Graduate School of Science, The University of Tokyo, Tokyo, Japan, 2 Mammalian Development Laboratory, Department of Gene Function and Phenomics, National Institute of Genetics, Mishima, Japan, 3 Advanced Genomics Center, National Institute of Genetics, Mishima, Japan, 4 Department of Genetics, The Graduate University for Advanced Studies, SOKENDAI, Mishima, Japan

* rajima@nig.ac.jp (RA); ysaga@nig.ac.jp (YS)

**Data Availability Statement:** • E8.5 embryo RNA-seq data have been deposited into the Gene Expression Omnibus (GEO) under accession code

## Abstract

The evolutionarily conserved RNA helicase DDX6 is a central player in post-transcriptional regulation, but its role during embryogenesis remains elusive. We here show that DDX6 enables proper cell lineage specification from pluripotent cells by analyzing *Ddx6* knockout (KO) mouse embryos and employing an *in vitro* epiblast-like cell (EpiLC) induction system. Our study unveils that DDX6 is an important BMP signaling regulator. Deletion of *Ddx6* causes the aberrant upregulation of the negative regulators of BMP signaling, which is accompanied by enhanced expression of *Nodal* and related genes. *Ddx6* KO pluripotent cells acquire higher pluripotency with a strong inclination toward neural lineage commitment. During gastrulation, abnormally expanded *Nodal* and *Eomes* expression in the primitive streak likely promotes endoderm cell fate specification while inhibiting mesoderm differentiation. We also genetically dissected major DDX6 pathways by generating *Dgcr8*, *Dcp2*, and *Eif4enif1* KO models in addition to *Ddx6* KO. We found that the miRNA pathway mutant *Dgcr8* KO phenocopies *Ddx6* KO, indicating that DDX6 mostly works along with the miRNA pathway during early development, whereas its P-body-related functions are dispensable. Therefore, we conclude that DDX6 prevents aberrant upregulation of BMP signaling inhibitors by participating in miRNA-mediated gene silencing processes. Overall, this study delineates how DDX6 affects the development of the three primary germ layers during early mouse embryogenesis and the underlying mechanism of DDX6 function.

## Author summary

Gene expression occurs through two steps: transcription (DNA to RNA) and translation (RNA to protein). Cells have highly sophisticated regulatory processes working on various levels for accurate gene expression. Post-transcriptional regulation, which includes all

GSE171156. • ESC & EpiLC RNA-seq data have been deposited into the Gene Expression Omnibus (GEO) under accession code GSE187390. All other data is available within the manuscript and Supporting Information files.

**Funding:** This work was supported by JSPS KAKENHI Grant Number 17H06166 to Y.S. https://www.jsps.go.jp/english/e-grants/ The funder did not play any role in the study design, data collection and analysis, decision to publish, or preparation of the manuscript.

**Competing interests:** The authors have declared that no competing interests exist.

RNA-related controls, is crucial because it enables fine-tuning and rapid alteration of gene expression. RNA-binding proteins and non-coding RNAs are the two main players in post-transcriptional regulation. DDX6, the subject of our study, is an RNA-binding protein, more specifically an RNA helicase, which can unwind or rearrange RNA secondary structures. Its diverse molecular and cellular functions have been reported, but its embryogenic role is unknown. Here, we describe DDX6 function during early mouse embryogenesis and the underlying mechanism using genetic methodology. DDX6 enables proper cell lineage specification of pluripotent stem cells by mainly regulating BMP signaling through miRNA-mediated gene silencing. As DDX6-mediated RNA regulation affects signaling pathways, the loss of DDX6 has a wide impact on developmental processes from pluripotency to embryo patterning. In addition to showing the developmental role of DDX6, we succeeded in the modular segregation of its various RNA-regulatory pathways. Considering the presence of DDX6 in diverse contexts, such as cancer, virus infection, and stem cells, this new knowledge forms a foundation for DDX6 being a good therapeutic target.

## Introduction

Post-transcriptional regulation, located in the middle layer of gene expression, is a critical controlling point where many mRNA regulatory processes occur. RNA-binding proteins (RBPs) and non-coding RNAs are the two main players [1,2]. Among diverse RBPs, RNA helicases are characterized by their wide range of involvement in RNA metabolism by binding RNA or remodeling ribonucleoprotein complexes (RNPs) [3,4]. DEAD box proteins compose the largest RNA helicase family sharing the Asp-Glu-Ala-Asp (DEAD) motif and having ATP-dependent RNA unwinding activity [5]. Mice have 43 DEAD box RNA helicases. Among them, we focused on DDX6 (Rck/p54 in human, Xp54 in *Xenopus*, Me31B in *Drosophila*, Cgh-1 in *C. elegans*, and Dhh1 in *S. cerevisiae*), an evolutionarily conserved RNA helicase throughout eukaryotes [6].

DDX6 participates in many aspects of RNA metabolism: processing body (P-body) formation [7,8], stress granule assembly, mRNA storage, mRNA decay [9], translational repression [10,11], microRNA (miRNA) pathway [12,13], and translational promotion [14,15]. P-bodies are membrane-less cytoplasmic mRNP granules thought to be the place of the storage or decay of translationally inactive mRNAs [13,16]. Many translational repression and decapping-related proteins, miRISC components, and translationally repressed mRNAs are accumulated in P-bodies. DDX6 is necessary for the formation and maintenance of P-bodies [17]. The helicase core of DDX6 and all other DEAD-box RNA helicases is composed of two RecA-like globular domains and they share conserved motifs [6]. The characteristic DEAD (Asp-Glu-Ala-Asp) sequence, which has helicase activity *in vitro* [18], is in Motif II. Motif VI is known to be associated with conformational changes coupled with ATP hydrolysis and unwinding activity [19]. DEAD helicase activity is necessary for P-body assembly in human iPSCs [20]. The tethering assay using only the RecA2 domain of *Xenopus* homologue p54 showed that this domain alone can repress the translation of target mRNAs without defects in P-body localization, but it was unable to form *de novo* P-bodies [8]. Another important finding was the functional defects of different helicase mutants were dependent on the changes in the interaction with other gene-repressing proteins. This is consistent with the notion that DDX6 acts as a partner protein of numerous RNA-regulatory proteins and a remodeler of mRNP complexes. As RNA helicases mostly interact with the sugar-phosphate backbone of RNAs, the target specificity

comes from cofactors [5,21,18]. Regarding its mRNA decay functions, DDX6 predominantly interacts with 5'-to-3' decay proteins [17]. In this direction of mRNA decay, decapping of the m7G cap structure is the rate-limiting step. Decapped mRNAs are finally degraded by the 5' monophosphate-dependent 5'-to-3' exoribonuclease, Xrn1 [22,23]. Although many molecular and cellular studies on DDX6 have been reported, investigation of its functions in embryonic development is limited.

Gastrulation is a milestone in embryogenesis because the primary germ layers that give rise to all cell types develop during this developmental event. The three germ layers of the embryo, the ectoderm, mesoderm, and endoderm, basically originate from the inner cell mass (ICM) of the blastocyst. Pluripotent embryonic stem cells (ESCs) can be derived from the ICM of E3.5 early blastocysts or the epiblast of E4.5 late blastocysts [24–27]. Cells at these stages are in the naive (ground) pluripotent state. Soon after implantation, naive epiblasts differentiate into the primed pluripotent state in which cells become capable of committing to a certain lineage during gastrulation [28–30]. The epiblast-like cells (EpiLCs) permit more precise staging of mouse pluripotent cells. Transcriptomic analysis demonstrated that EpiLCs, induced from ESCs, have similar properties to the epiblast of post-implanted, pre-gastrulating (E5.5~6.0) embryos [31]. These *in vitro* systems enable detailed examination of pre- and early post-implantation embryos, which are normally difficult to investigate *in vivo* due to their small size.

From the 8-cell stage, transcription factors and signaling pathways play a major role in cell fate determination [32]. The transforming growth factor-β (TGF-β) superfamily is one of the major signaling pathways involved in early mammalian development. Bone morphogenetic protein (BMP) and Nodal belong to different subgroups, and they utilize distinctive serine/threonine kinase receptors for signal transduction. Activated receptors phosphorylate downstream intracellular mediators, receptor SMADs 1, 5, and 8 for BMP signaling, and SMADs 2 and 3 for Nodal signaling [33,34]. BMP and Nodal signaling are known to mutually antagonize each other, and this antagonism is generally thought to occur through competition for the common signal mediator SMAD4 [35–38]. BMP signaling has multiple roles during early post-implantation development. It is required for extra-embryonic mesoderm formation [39], primordial germ cell (PGC) induction [40], mesoderm development, and patterning [41], and inhibiting premature neural differentiation [42]. BMP signaling is dispensable for ESC self-renewal but is required for proper differentiation [43,44]. The Nodal pathway exerts its influence when ESCs transition from naive pluripotency. Nodal is important for securing primed pluripotency with the capacity to differentiate into multiple lineages [30].

There are a few studies that examined the developmental roles of DDX6 *in vivo*. DDX6 function is crucial for maintaining adult spermatogonial stem cells [45] and for helping NANOS2 to induce male-type germ cell differentiation in embryonic testes [46]. During female germ cell development, deletion of *Ddx6* causes defects in primordial follicle formation [47]. Two previous studies investigated the role of DDX6 in mouse and human pluripotent cells. In mESCs, DDX6 is necessary for translational repression of miRNA targets and *Ddx6* knockout (KO) cells exhibited similar phenotypes to *Dgcr8* KO ESCs, which lack all miRNAs [48]. Another study elucidated the relationship between stem cell potency and P-body-mediated translational regulation. As an essential factor of P-body formation, once DDX6 was depleted, P-bodies were disassembled and translationally suppressed target mRNAs, including many transcription factors and chromatin regulators, re-entered the translation pool. The resulting increased expression of target genes altered chromatin organization and made both human and mouse primed ESCs become more naively pluripotent and resistant to differentiation [20].

However, these examinations were conducted on ESCs or very limited cell types. To deepen our understanding of the role of DDX6 as a key post-transcriptional regulator during early

mouse embryogenesis, we examined *Ddx6* KO embryos. Gastrulation stages were investigated in embryos and the earlier time points were assessed using an ESC-to-EpiLC induction model. This study revealed that DDX6 exerts potent effects on the development of the three primary germ layers by preventing aberrant inhibition of the BMP signaling pathway. Furthermore, through genetic dissection of the DDX6 pathways, we found that DDX6 works through the miRNA pathway, but P-bodies are dispensable during early development.

## Results

### *Ddx6* knockout results in embryonic lethality with severe morphological defects

Before investigating the functions of DDX6 during embryogenesis, we examined its expression pattern through DDX6 immunohistochemistry (IHC). In embryonic day (E) 6.5 embryos, DDX6 was highly and ubiquitously expressed, localizing in cytoplasmic foci (S1A Fig). All DDX6 foci co-localized with DCP1A foci, a P-body-specific marker, indicating that DDX6 is expressed in P-bodies. At E7.5, DDX6 expression was strongest in the epiblast and there were many distinct P-body foci (S1B Fig). BRACHYURY-positive emerging and migrating mesoderm cells had relatively weaker DDX6 expression with a fewer number of P-bodies. In E8.5 embryos, DDX6 expression was observed in all areas, including the neuroepithelium, tailbud, and somites (S1C Fig). DDX6 was ubiquitously expressed in early embryos in forming P-bodies.

To clarify the role of DDX6 in embryonic development, we generated a $Ddx6^{\triangle/+}$ mouse line and crossed heterozygous mice to get $Ddx6^{\triangle/\triangle}$ (KO) embryos. The genotyping results of the collected embryos are displayed in Table 1. No $Ddx6^{\triangle/\triangle}$ embryo was found from E11.5. The proportion of homozygous KO embryos at E7.5~E8.5 (12.8% and 15.4%) was already lower than the expected Mendelian ratio, which signifies that some mutant embryos died even earlier than E7.5.

Developmental defects were observed from E6.5 (Fig 1). At E7.5, mutants were smaller, but an extraembryonic body part developed and embryos formed a cylindrical shape. E8.5 *Ddx6* KO embryos exhibited developmental delay and posterior defects with some variability (S1D Fig). They were broadly divided into two embryonic stages: before (Panel 1 in Fig 1 and Panels 1–3 in S1D Fig) and after (Panel 2 in Fig 1 and Panels 4–6 in S1D Fig) head-fold formation. The variance in posterior defects was also observed in E9.5 mutants. Some developed a mid-posterior body (Fig 1 and Panel 1 in S1E Fig), whereas others exhibited marked posterior truncation (Panels 2–3 in S1E Fig). In conclusion, DDX6 is necessary for mouse embryonic development.

**Table 1. *Ddx6* knockout results in embryonic lethality by E11.5.**

| Time point | +/+ | △/+ | △/△ | Total |
|:---:|:---:|:---:|:---:|:---:|
| E6.5 | 6 (19.4%) | 16 (51.6%) | 9 (29%) | 31 |
| E7.5 | 61 (37.2%) | 82 (50%) | 21 (12.8%) | 164 |
| E8.5 | 81 (30.3%) | 145 (54.3%) | 41 (15.4%) | 267 |
| E9.5 | 17 (23.3%) | 43 (58.9%) | 13 (17.8%) | 73 |
| E10.5 | 12 (42.9%) | 11 (39.3%) | 5 (17.9%) | 28 |
| E11.5 | 11 (44%) | 14 (56%) | 0 | 25 |
| E12.5 | 7 (30%) | 16 (70%) | 0 | 23 |

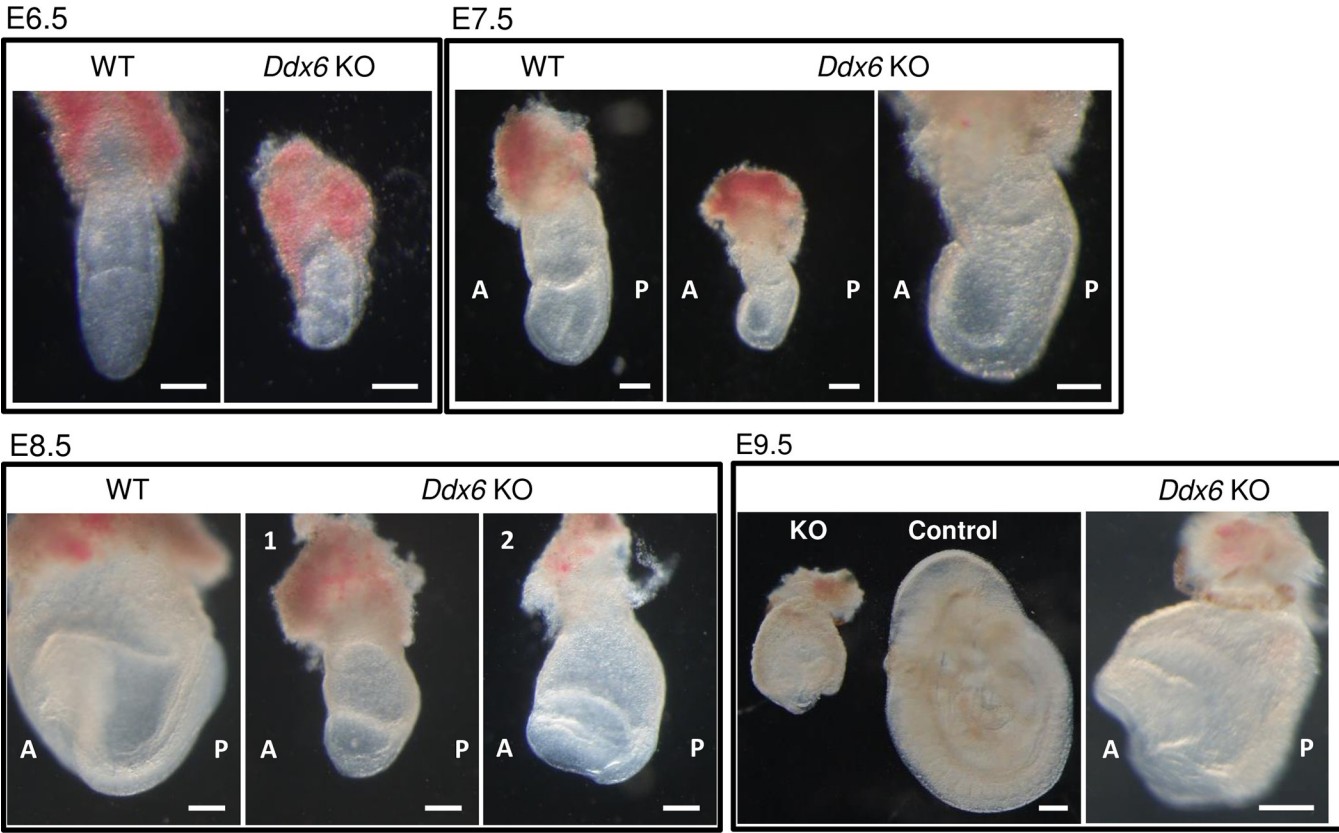

**Fig 1. $Ddx6^{\triangle/\triangle}$ embryos exhibited growth delay and morphological defects.** Pictures of embryos prepared from E6.5-E9.5 littermates. At E8.5, two morphologically distinct mutant embryos are shown; earlier than head-fold formation (1) and with head-fold structure (2). The right panels of E7.5 and E9.5 are the magnified images of $Ddx6^{\triangle/\triangle}$ embryos in the left panels. (Scale: 100 μm for E6.5 and E7.5; 200 μm for E8.5 and E9.5).

## $Ddx6^{\triangle/\triangle}$ embryos display phenotypes arising from the disrupted BMP signaling pathway

To characterize *Ddx6* mutant embryos at the transcript-level and further find the possible causes of defects, we performed RNA sequencing (RNA-seq). Two E8.5 *Ddx6* KO cDNA libraries (KO1 and KO2) were generated. KO1 comprised one Panel 1 in Fig 1-like embryo and one Panel 2 in Fig 1-like embryo, whereas KO2 was composed of three Panel 1 in Fig 1-like embryos. As there was variability between the two samples, we concluded them to be insufficient to provide statistical significance and decided to utilize RNA-seq data just to provide a rough trend and guide subsequent experiments. Gene ontology (GO) term enrichment analysis indicated that genes of major developmental processes, especially cell fate commitment and the formation of mesoderm derivatives, such as muscle tissue development, muscle organ morphogenesis, cardiac myofibril assembly, skeletal muscle organ development, and connective tissue development, were downregulated in *Ddx6* KO libraries (S2A Fig). In contrast, the terms associated with cell death, immune response, cell metabolism, and negative regulation of BMP signaling pathway-related genes were highly upregulated. Negative regulation of the BMP signaling pathway was notable because BMP signaling has multiple important roles during early embryogenesis. Several BMP negative regulators were upregulated in $Ddx6^{\triangle/\triangle}$ embryos (S2B Fig). Based on the reported functions, genes that are listed as the negative regulators of BMP signaling were classified into five clusters: receptor-related (*Inhbb*, and

*Tmprss6*), secreted BMP antagonists (*Cer1*, *Chrd*, *Chrdl2*, *Noggin*, *Grem2*, and *Htra1*), TGF-β signaling-related (*Lgals9*, *Xdh*, *Pai1*, *Hpgd*, and *miR-382*), FGF signaling-related (*Fgf5*), and intracellular inhibitor (*Nanog*) (S2C Fig). As genes related to the inhibition of BMP signaling were highly upregulated, we examined whether BMP signaling is repressed in *Ddx6*$^{\triangle/\triangle}$ embryos. We reasoned that if BMP signal transduction is indeed dysfunctional, then *Ddx6*$^{\triangle/\triangle}$ embryos should exhibit the representative phenotypes of BMP signaling mutant embryos.

### 1) Mesoderm formation defects

BMP signaling is required for mesoderm formation and posterior body development [41,49]. Whole-mount *in situ* hybridization (WISH) with a *Brachyury (T)* probe showed that the primitive streak existed in E8.5 *Ddx6*$^{\triangle/\triangle}$ but it did not extend anteriorly (Figs 2A and S1F). The somites were barely developed. We also examined the induction timepoint of *Brachyury*. Gastrulation begins around E6.5 as the primitive streak forms and is preceded by *Brachyury* expression [50]. Unlike WT, *Ddx6*$^{\triangle/\triangle}$ embryos started expressing BRACHYURY from E7.5 (Fig 2B). E7.5 WISH also revealed a small amount of *Brachyury* in the very proximal posterior region of the embryo (Fig 2A). Visualization of *Otx2* expression, which marks the head region, indicated the lack of posterior body in mutants, which have not formed head fold yet (Fig 2C). In summary, there was a delay of primitive streak formation in *Ddx6*$^{\triangle/\triangle}$ embryos, and their shortened and widened primitive streak (S1F Fig) suggested that the nascent mesoderm population has defects in differentiation and subsequent ingression. *Ddx6*$^{\triangle/\triangle}$ embryos had defects in posterior body development and mesoderm differentiation similar to BMP signaling mutant embryos.

### 2) Premature neural induction

BMP signaling also prevents premature neural induction [42]. We examined whether this function was also impaired in *Ddx6*$^{\triangle/\triangle}$ embryos. SOX1 is the earliest neuroectoderm marker and is normally not detected until E7.5 in WT embryos [51,42]. However, E6.5 *Ddx6*$^{\triangle/\triangle}$ embryos exhibited clear SOX1 expression in all epiblast cells (Fig 2D). Additionally, premature neuronal differentiation was detected in *Ddx6*$^{\triangle/\triangle}$ based on RNA-seq. The markers of neural stem cells (NSCs) and neural progenitor cells (NPCs), such as *Sox1* and *Pax6*, were downregulated (S2D Fig), but genes of neuron-restricted progenitors and differentiated post-mitotic neuronal cells were upregulated (S2E Fig). Section IHC confirmed that protein expression resembled transcript levels (Fig 2E). The earliest neuroectoderm marker, SOX1, and a persistent marker of NSC and NPC, SOX2 [52], were weakly expressed in E8.5 *Ddx6*$^{\triangle/\triangle}$ embryos. In summary, in *Ddx6*$^{\triangle/\triangle}$ embryos, the neural lineage was precociously induced as in BMP receptor mutant embryos [42]. Moreover, *Ddx6*-deficient NSCs showed defective self-renewal maintenance and prematurely differentiated.

### *Ddx6*$^{\triangle/\triangle}$ embryos exhibit defects that are related to the enhanced *Nodal* expression

Another indication of suppressed BMP signaling in *Ddx6*$^{\triangle/\triangle}$ embryos was increased *Nodal* expression (S2F Fig). BMP and Nodal signaling are often in a competitive relationship and can suppress each other. During gastrulation, a gradient of NODAL activity patterns the primitive streak and allocates mesendoderm progenitors. NODAL and its downstream target EOMES together define the anterior primitive streak (APS), from which cardiac mesoderm and definitive endoderm progenitors are specified [53–56]. WISH demonstrated that *Nodal* expression was confined to the node in E7.5 WT embryos, but its expression was highly spread over the proximal-posterior region in *Ddx6*$^{\triangle/\triangle}$ embryos (Fig 3A). The expression was eventually

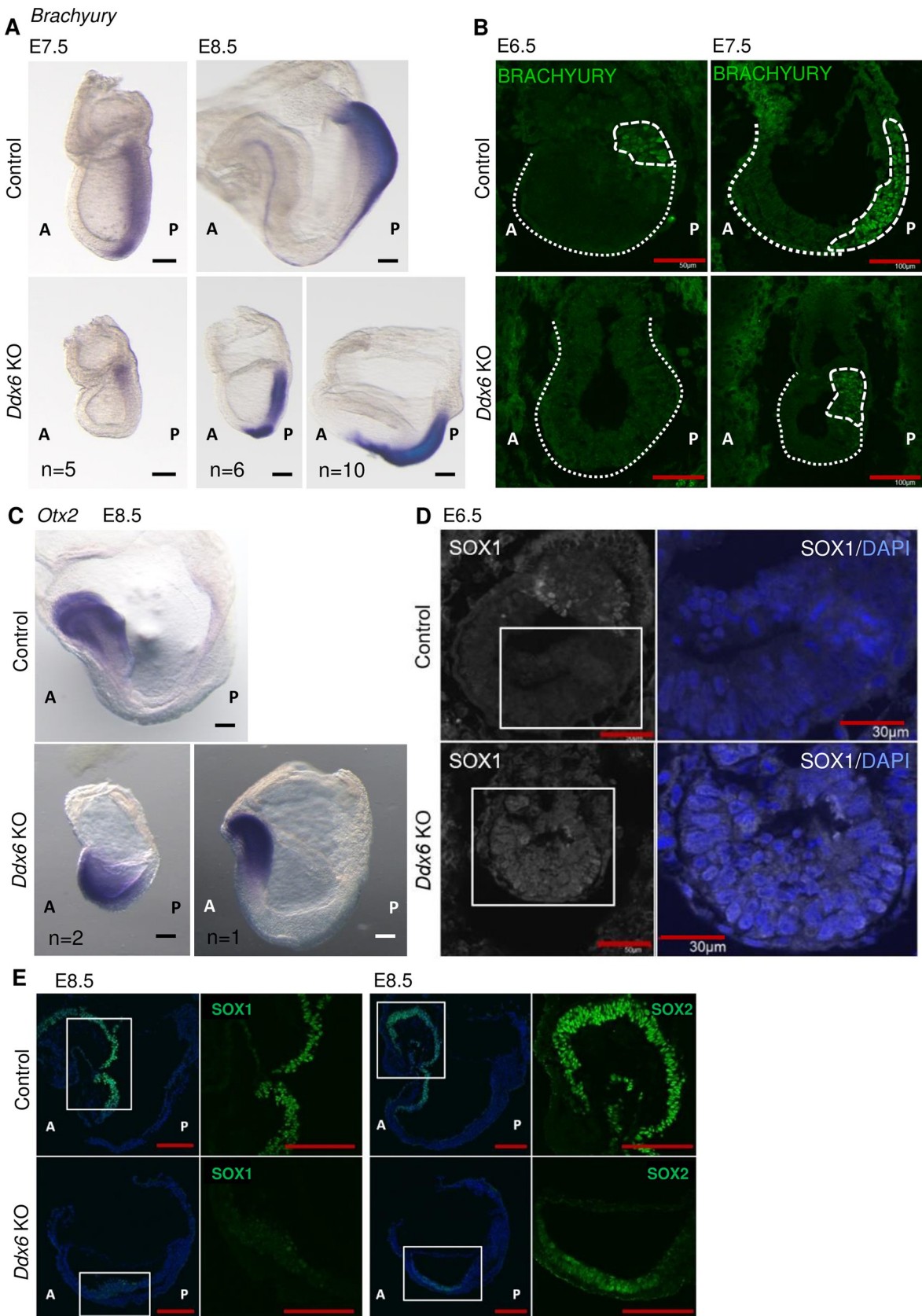

**Fig 2. $Ddx6^{\triangle/\triangle}$ embryos display developmental defects: mesoderm formation failure and premature neural induction.** (A) Whole-mount ISH of E7.5 & E8.5 embryos with a *Brachyury* probe (Scale: 100 μm, n = 5 for E7.5; 200 μm, n = 16 for E8.5). (B) E6.5 & E7.5 embryo frozen section IHC for BRACHYURY (Scale: 50 μm for E6.5, n = 2; 100 μm for E7.5, n = 3). Embryo parts are indicated by dotted lines. (C) Whole-mount ISH of E8.5 embryos with an *Otx2* probe (Scale: 100 μm, n = 3). (D) E6.5 embryo frozen section IHC for SOX1 (Scale: 50 μm for lower magnification, 30 μm for higher magnification, n = 2). Square parts are enlarged in the right panels. Shown with DAPI staining. (E) E8.5 embryo frozen section IHC for SOX1 & SOX2 (Scale: 100 μm, n = 3).

restricted to the node at E8.5 in mutants in the head fold stage, but one mutant without a head fold retained high *Nodal* expression. *Eomes* is highly expressed in the extraembryonic ecto-derm and the posterior part of the epiblast at E6.5. Its expression moves distally to the primi-tive streak at E7.5 [57], and is reduced by E8.5 in the WT embryos. However, some E8.5 $Ddx6^{\triangle/\triangle}$ embryos retained high-level expression in the primitive streak (Fig 3B). The expres-sion pattern in E7.5 KO embryos resembled that of earlier stage (~E6.5) control embryos. Some E8.5 $Ddx6^{\triangle/\triangle}$ embryos were morphologically similar to the control E7.5 embryos, but expression levels of *Nodal* and *Eomes* were much stronger, suggesting that these genes are highly upregulated regardless of certain embryonic stages. RNA-seq analyses showed that the KO2 sample had downregulated expression of differentiated mesoderm genes, whereas the expression of endoderm lineage genes was upregulated (S2G Fig). KO2 exhibited higher expression of mesendoderm progenitor markers (*Mixl1*, and *Gsc*), endoderm progenitor marker (*Lhx1*), and definitive endoderm markers (*Sox17*, and *Foxa2*). The 'Endoderm cell fate specification' category was also enriched in GO term analysis of most upregulated genes in $Ddx6^{\triangle/\triangle}$ (S2A Fig). Therefore, posteriorly expanded high expression of the APS marker *Nodal* and *Eomes* disturbed the patterning of the primitive streak, which likely directed mesen-doderm progenitors toward the endodermal lineage. Transcript levels of some key genes were further assessed by qRT-PCR using separately prepared $Ddx6^{\triangle/\triangle}$ embryos and the results were consistent with the RNA-seq data (S2H Fig).

Enhanced *Nodal* expression also affected pluripotency. Nodal signaling is important for regulating primed pluripotency. Activin/Nodal signaling is necessary to induce *Nanog* tran-scription in mEpiSCs [58]. NODAL is also required to activate and maintain *Nanog* expression in the proximal epiblast of peri-gastrula mouse embryos [59]. Therefore, we examined whether the expression of this other NODAL target gene also increased in $Ddx6^{\triangle/\triangle}$ embryos. The core pluripotency factors *Nanog* and *Pou5f1* (*Oct4*) and naive pluripotency marker *Klf4* were upre-gulated in $Ddx6^{\triangle/\triangle}$ (S2I Fig). The retained expression of the naive pluripotency-specific genes in E8.5 embryos suggested that exit from the ground pluripotent state did not occur properly in *Ddx6* mutants. We examined NANOG expression by section IHC (Fig 3C). In WT embryos, NANOG expression was strongest in the primitive streak at E6.5, but at E7.5, its expression moved anteriorly, and the posterior part was negative. Contrarily, E7.5 $Ddx6^{\triangle/\triangle}$ embryos exhibited strong NANOG expression in the posterior epiblast and this expression was main-tained at E8.5.

Of note, the expression pattern of TuJ1 (TUBB3) was abnormal, being another indication of the failure of $Ddx6^{\triangle/\triangle}$ embryos exiting from the pluripotent state (Fig 3D). In E6.5 WT embryos, low but positive TuJ1 expression was detected in the epiblast and its expression increased by E7.5. In E7.5 embryos, T-positive mesendoderm progenitors retained TuJ1 expression, but TuJ1 expression was eventually turned off with the progression of differentia-tion and only remained in the neuroepithelium. This suggests that a well-known marker of neuroectoderm and neurons, TuJ1, is also expressed in the primed pluripotent epiblast. In *Ddx6* mutants, the expression level of TuJ1 was much higher than in WT at all time points. The stronger expression in *Ddx6* KOs may have resulted from the promoted pluripotency and premature neuroectoderm differentiation.

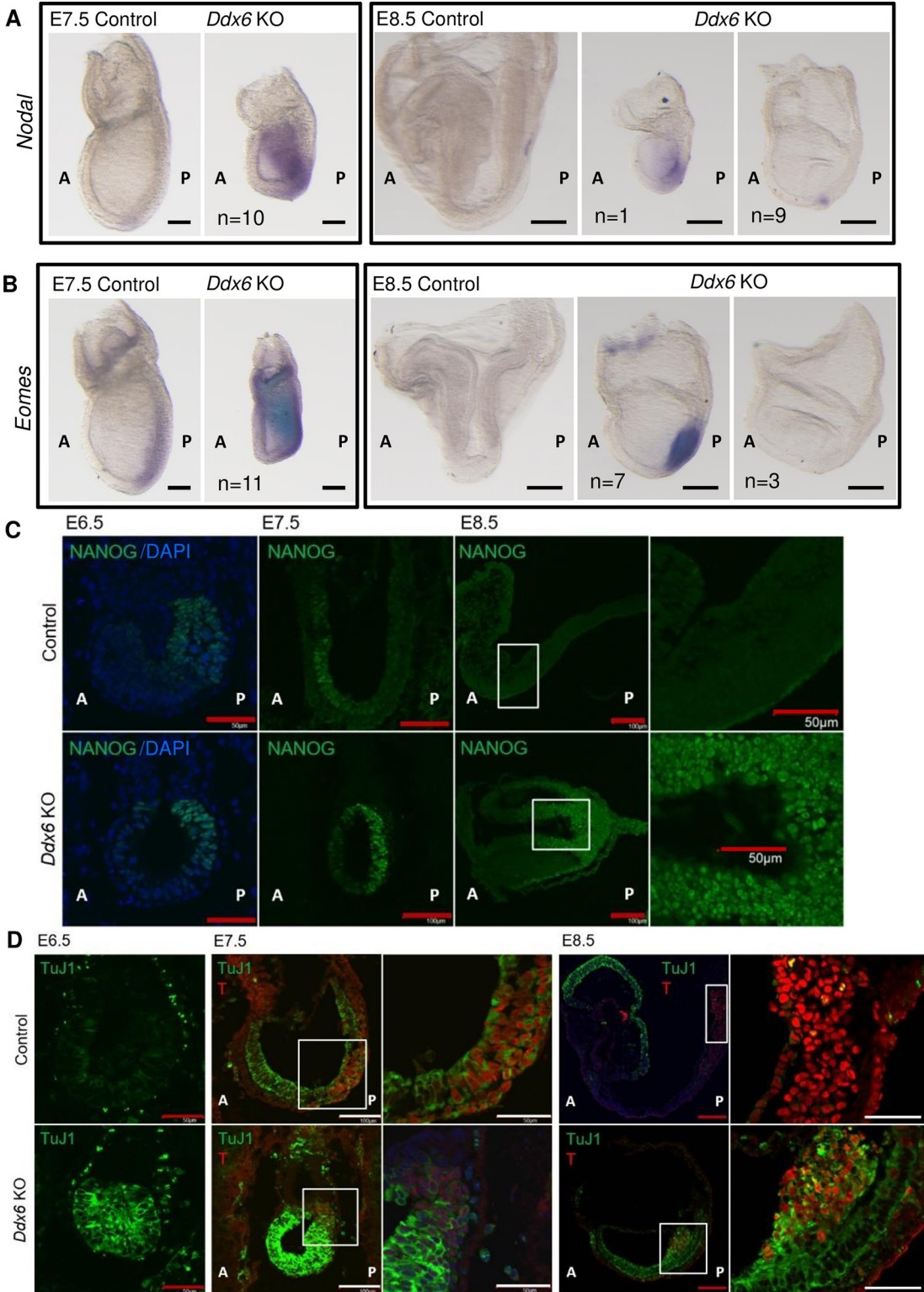

**Fig 3. *Ddx6*<sup>△/△</sup> embryos exhibit defects that are associated with increased *Nodal* expression.** (A-B) Whole-mount ISH of E7.5 & E8.5 embryos with (A) a *Nodal* probe (Scale: 100 μm, n = 10 for E7.5; 200 μm, n = 10 for E8.5) and (B) an *Eomes* probe (Scale: 100 μm, n = 11 for E7.5; 200 μm, n = 10 for E8.5). Expressions of *Nodal* and *Eomes* were maintained in E8.5 *Ddx6* KO embryos, which were younger than the head-fold stage (left), but down-regulated as in WT at the head-fold stage (right). (C) E6.5~E8.5 embryo frozen section IHC for NANOG (Scale: 50 μm for E6.5, n = 2; 100 μm for E7.5, n = 3; 100 μm for lower magnification of

E8.5, 50 μm for higher magnification, n = 3). Square parts are enlarged in the right panels. (D) TuJ1 expression in the epiblast. E6.5~E8.5 embryo frozen section IHC for TuJ1 & T (BRACHYURY). The signal intensity is comparable only between the same embryonic day samples. (Scale: 50 μm for E6.5, n = 2; 100 μm for lower magnification, 50 μm for higher magnification, n = 3 for E7.5 & E8.5). Square parts are enlarged in the right panels.

E8.5 $Ddx6^{\triangle/\triangle}$ embryos retained ectopic and high expression of NANOG and TuJ1 in the posterior part, indicating that posterior epiblast cells failed to turn off pluripotent gene expression. This aberrant state likely interrupted the differentiation of mesendoderm-committed cells. Enriched GO terms among downregulated genes from RNA-seq (S2A Fig), such as 'Cell fate commitment' and 'Cell fate determination', supported the strengthened and sustained pluripotency of $Ddx6^{\triangle/\triangle}$ embryos.

## $Ddx6^{\triangle/\triangle}$ pluripotent cells also show repressed BMP signaling with enhanced *Nodal* expression

We examined E8.5 $Ddx6^{\triangle/\triangle}$ embryos and considered repressed BMP signaling as a major cause of their developmental defects. We then looked for the earliest time point of inhibited BMP signaling in $Ddx6^{\triangle/\triangle}$. There were no morphological abnormalities until E3.5 blastocysts and ESCs were successfully established. DDX6 was highly expressed in ESCs and EpiLCs in P-bodies (S3A Fig), which were disassembled in $Ddx6^{\triangle/\triangle}$ cells (S3B Fig). The proliferation rate of $Ddx6^{\triangle/\triangle}$ ESCs was lower (Fig 4A), but they had no defects in maintaining pluripotency over many passages. However, like E8.5 $Ddx6^{\triangle/\triangle}$ embryos, they expressed higher levels of pluripotency genes, such as *Oct4*, *Nanog*, and *Sox2*, than WT ESCs (Fig 4B).

As $Ddx6^{\triangle/\triangle}$ blastocysts and ESCs did not exhibit notable abnormalities, we examined their next developmental capacity. We conducted ESC-to-EpiLC induction to mimic natural *in vivo* development. The transition of the naive ground state ESCs to the primed pluripotent state EpiLCs takes two days and EpiLC Day1 is regarded as a transition state that exhibits a distinctive open chromatin landscape and transcriptome [60]. During EpiLC induction, the difference in cell number between WT and $Ddx6^{\triangle/\triangle}$ cells markedly increased (Fig 4C). We then investigated the expression pattern of several key genes during EpiLC induction by qRT-PCR. First, we checked well-known epiblast markers *Fgf5* and *Nodal*, and an important regulator of the transition state, *Zic3*, which exhibits peak expression on EpiLC Day1 [60]. $Ddx6^{\triangle/\triangle}$ cells had significantly higher expression of *Nodal* and *Zic3*, but there was no difference in *Fgf5* expression (Fig 4D). After confirming induction, we examined the expression profile of pluripotency and early differentiation-related genes. As shown in Fig 4B, $Ddx6^{\triangle/\triangle}$ ESCs exhibited slightly higher expression of pluripotency genes and this difference increased during EpiLC induction (Fig 4E). They had much higher expression of naive pluripotency markers (*Klf4*, *Rex1*), suggesting that even though an overall transition was made to the EpiLC state, cells failed to completely exit from the ground state. We also noted a change in the differentiation capacity of $Ddx6^{\triangle/\triangle}$ cells. Neural lineage-inducing genes, such as *Sox1*, *Sox2*, and *Pax6*, were highly upregulated in $Ddx6^{\triangle/\triangle}$ cells, whereas the mesendoderm lineage inducer *T* was significantly downregulated (Fig 4F). We also conducted a monolayer differentiation experiment. ESCs favor neuronal differentiation in low-density, serum-free, and feeder-free culture conditions [61]. Compared with WT ESCs, $Ddx6^{\triangle/\triangle}$ ESCs differentiated and developed into neurons quickly. $Ddx6^{\triangle/\triangle}$ cells exhibited stronger expression of TuJ1 with the morphology of well-developed dendrites and axons on differentiation Day1 (Fig 4G). This was similar to the premature neural differentiation observed in E8.5 $Ddx6^{\triangle/\triangle}$ embryos. Taken together, $Ddx6^{\triangle/\triangle}$ embryos developed normally until the blastocyst stage and ESCs had no defects in self-renewal. However, the differentiation capacity of $Ddx6^{\triangle/\triangle}$ pluripotent cells was strongly skewed to neuronal lineage commitment.

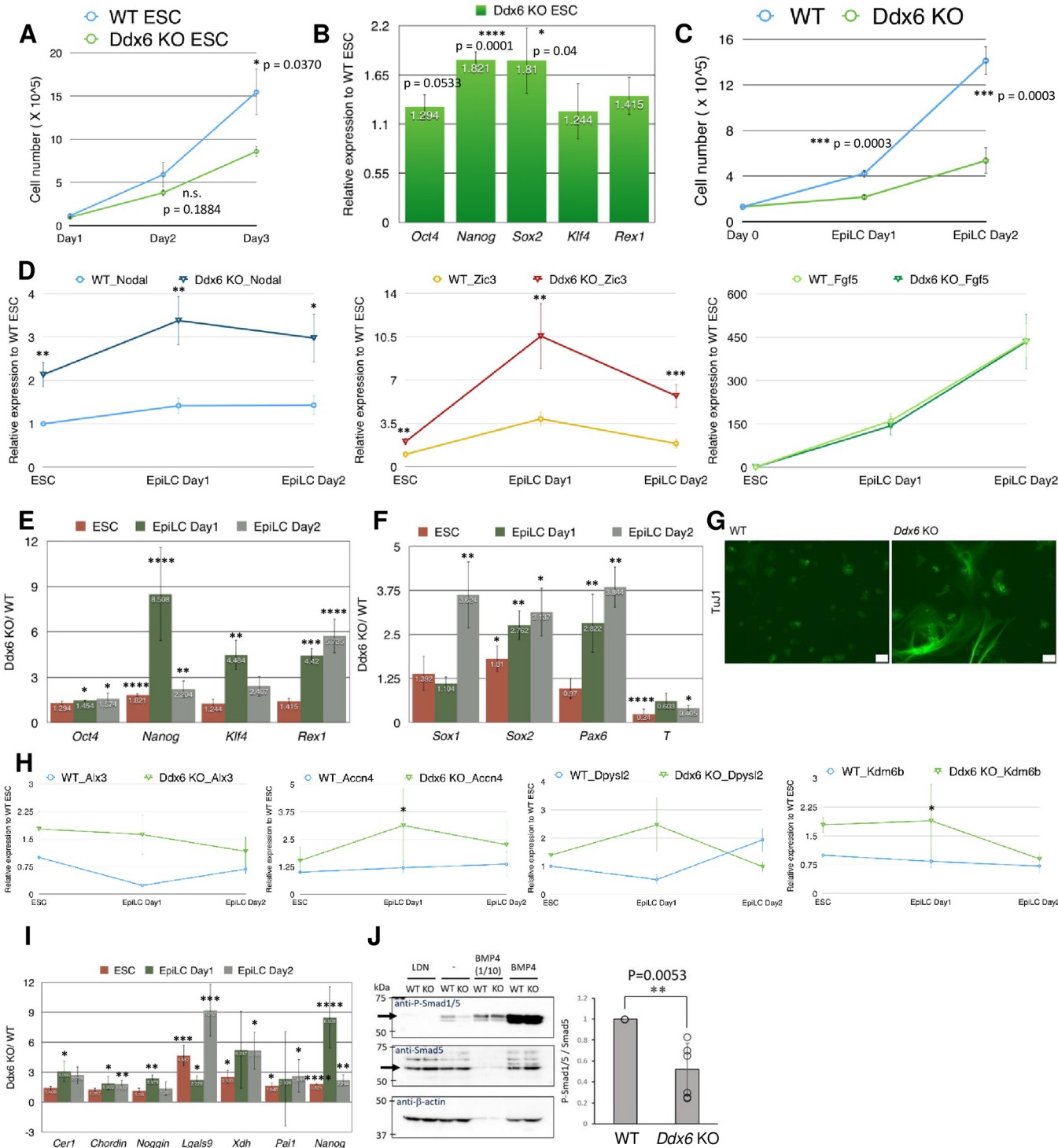

**Fig 4. BMP signaling is repressed in *Ddx6*$^{\triangle/\triangle}$ pluripotent cells.** (A) Cell counting of ESCs over a three-day culture period. Mean ± SEM. Significance was calculated by the Student's t-test (n = 5). (B) qRT-PCR examining the relative expression of pluripotency markers in *Ddx6*$^{\triangle/\triangle}$ ESCs to WT ESCs. Mean ± SEM. Student's t-test (n = 7~9). (C) Cell counting during the ESC-to-EpiLC induction period. Mean ± SEM. Student's t-test (n = 13 for WT, n = 7 for *Ddx6*$^{\triangle/\triangle}$). (D) qRT-qPCR examining the expression pattern of *Nodal*, *Fgf5*, and *Zic3*. Mean ± SEM. Student's t-test (n = 9 for *Nodal* & *Fgf5*, n = 7 for *Zic3*) (*p ≤ 0.05, **p ≤ 0.01, ***p ≤ 0.001, ****p ≤ 0.0001). (E-F, I) qRT-PCR analysis of the expression trend of several key genes during the EpiLC induction period. Each bar represents the relative expression of *Ddx6*$^{\triangle/\triangle}$ cells to WT cells at the indicated time point. Mean ± SEM. Student's t-test. (E) Major pluripotency genes (n = 7~9). (F) Early neuroectoderm and mesendoderm lineage markers (n = 7~9). (I) The negative regulators of BMP signaling (n = 6~9). (G) TuJ1 ICC (immunocytochemistry) on Day1 of monolayer differentiation (Scale: 50 µm) (n = 3). (H) qRT-PCR analysis of the expression of the known BMP-SMAD1/5

target genes in ESCs. Each value is the relative expression to WT ESCs (n = 5). (J) Western blot for endogenous SMAD5 and phosphorylated SMAD1/5 in mESCs. Four conditions: LDN (LDN193189: BMP Type I receptor ALK2/3 inhibitor)/—(non-treated)/ BMP4 1/10 (1/10 dilution of rBMP4 treated sample)/ BMP4 (rBMP4 treated). The quantified signal intensity of the band is displayed on the right-side graph (n = 6).

BMP signaling is important to prevent the differentiation of ESCs to the neuronal lineage [62]. Thus, the strong inclination of $Ddx6^{\triangle/\triangle}$ ESCs toward the neuronal cell fate suggests that this brake is nonfunctional. We investigated whether the SMAD-dependent BMP signaling pathway was repressed in $Ddx6^{\triangle/\triangle}$ cells. In ESCs and the initial differentiation stage, BMP signaling has a transcriptional repressive role for these SMAD1/5 target genes [43]. We examined the expression pattern of known SMAD1/5 target genes in ESCs, and found that the expression of *Accn4*, *Alx3*, *Dpysl2*, and *Kdm6b* was higher in $Ddx6^{\triangle/\triangle}$ cells (Fig 4H). This suggested that target genes were transcriptionally de-repressed due to reduced BMP signaling. In particular, this characteristic higher expression was prominent on EpiLC Day1, which corresponds to the earliest differentiation state. DPYSL2 and the H3K27 demethylase KDM6B are early neural differentiation regulators [63,43]; therefore, their high expression is consistent with the phenotype of $Ddx6^{\triangle/\triangle}$ ESCs preferring neural lineage commitment. We next investigated whether aberrant upregulation of a set of BMP signaling inhibitors also occurred in $Ddx6^{\triangle/\triangle}$ pluripotent cells. Like $Ddx6^{\triangle/\triangle}$ embryos, $Ddx6^{\triangle/\triangle}$ ESCs exhibited a significant increase in the expression of negative regulators of the BMP pathway during EpiLC induction (Fig 4I). Lastly, we confirmed that this increased expression of BMP inhibitors indeed reduced BMP signaling via Western analysis (Fig 4J). The p-SMAD1/5 protein amount was significantly reduced in $Ddx6^{\triangle/\triangle}$ ESCs. There was no difference in SMAD5 protein amount, and transcript levels of the BMP signaling components, such as ligands, receptors, and Smad mediators, were similar (S2J Fig). Along with suppressed BMP signaling, $Ddx6^{\triangle/\triangle}$ ESCs displayed high expression levels of *Nodal* and *Nanog*, analogous to $Ddx6^{\triangle/\triangle}$ embryos (Fig 4D and 4E). We hypothesized that Nodal signaling was promoted in $Ddx6^{\triangle/\triangle}$ ESCs, but we did not observe a significant difference in the amount of p-SMAD2 (S4A Fig). In conclusion, the common features of $Ddx6^{\triangle/\triangle}$ ESCs and embryos suggest that suppression of BMP signaling with an increased *Nodal* expression occurs when DDX6 is absent.

## Depletion of DDX6 quickly induces transcriptional upregulation of the negative regulators of BMP signaling and *Nodal*

To further ask whether the aberrant activation of BMP signaling inhibition is a primary property of $Ddx6^{\triangle/\triangle}$ cells, we conditionally deleted *Ddx6* using the *Rosa-CreER^{T2}; Ddx6^{flox/flox}* mouse line. As we expected the loss of DDX6 to cause mesoderm formation defects, we removed DDX6 during gastrulation. We first examined the time required for the complete depletion of DDX6. When tamoxifen was administered to the pregnant female via oral gavage at E6.5, *Ddx6* deletion and depletion of existing DDX6 proteins were completed by E7.5 (Fig 5A). We then injected tamoxifen at E6.5 and collected embryos at E8.5 to examine their phenotypes. Conditional knockout (cKO) embryos of the same litter exhibited variable phenotypes like conventional KO embryos (Fig 5B). A few had marked posterior truncation (KO2, KO17). The others developed the mid-to-posterior body part, but it was shorter and smaller, and the head and heart were abnormal (KO7, KO12). Although the phenotypes were milder than those of conventional KO embryos, conditional KO embryos also demonstrated characteristic gene expression of *Ddx6* mutants. The expression of the negative regulators of BMP (*Chrd*, *Noggin*, *Lgals9*, *Xdh*, *Pai*), *Nodal*, and *Eomes* increased after the depletion of DDX6 (Fig 5C). Therefore, along with the features of $Ddx6^{\triangle/\triangle}$ ESCs, this conditional KO experiment revealed that these characteristic gene expression changes are closely related to the absence of

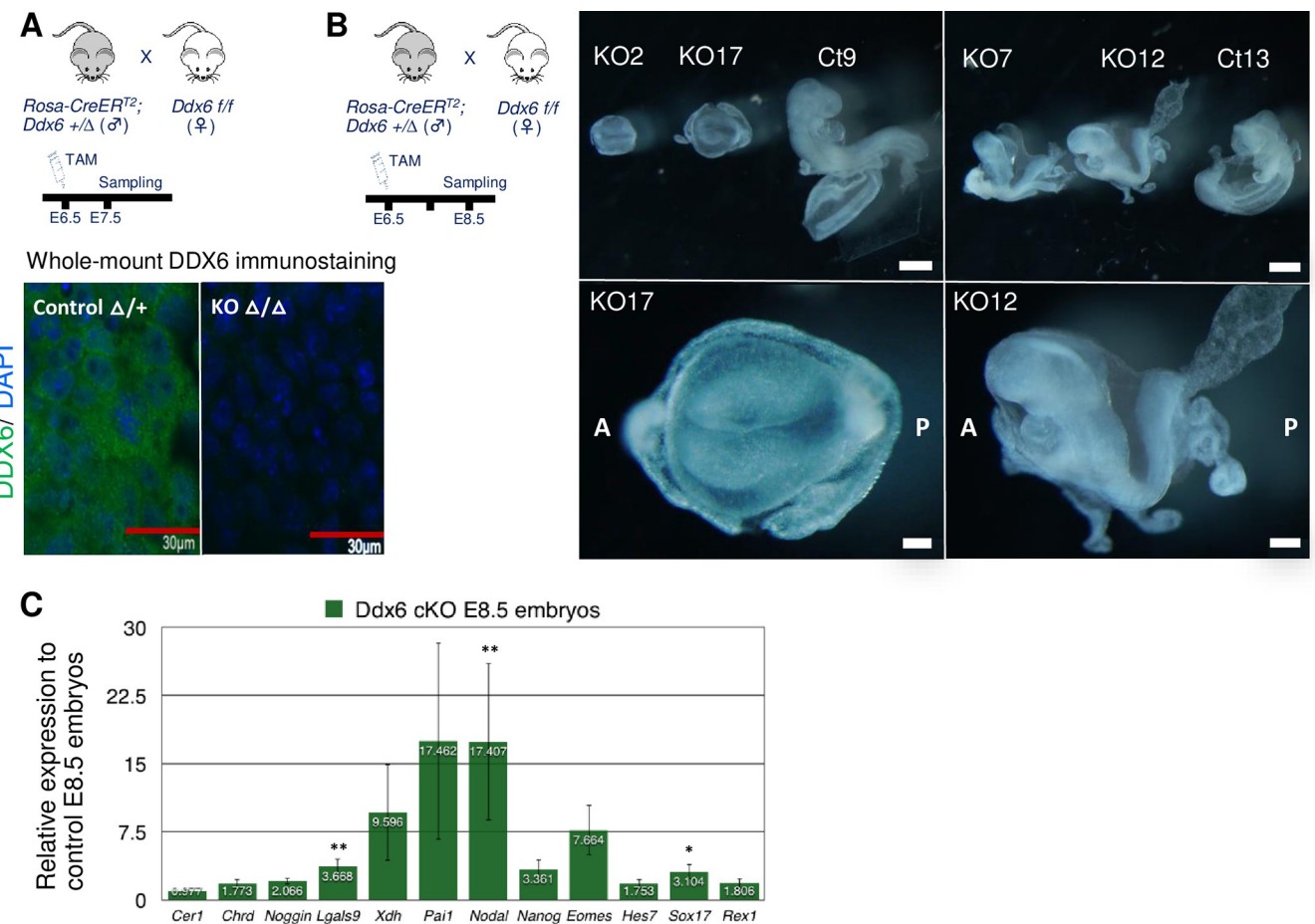

**Fig 5. Conditional knockout of *Ddx6* quickly upregulates expression of the BMP signaling inhibitors and *Nodal*.** (A) Tamoxifen was injected at E6.5. Whole-mount DDX6 immunostaining confirmed that the complete depletion of DDX6 takes approximately 1 day (Scale: 30 μm, n = 3). DDX6 in green, DAPI in blue. (B) E8.5 cKO embryos exhibited similar phenotypes to conventional KO embryos (Scale: 500 μm for group, 100 μm for KO17, 200 μm for KO12) (n = 6). (C) qRT-PCR analysis of several key genes in *Ddx6* cKO E8.5 embryos. Embryos exhibiting similar morphology to the KO12 were used for analysis. Mean ± SEM. Significance was calculated by the Wilcoxon rank-sum test (n = 9~10) (* $\alpha$ = 0.05 significance level, ** $\alpha$ = 0.01).

DDX6 rather than the cumulative result of indirect effects. It also reconfirmed that DDX6 is essential during gastrulation and that the loss of DDX6 makes cells activate inhibitory regulation of BMP signaling.

## Genetic dissection of the DDX6-mediated RNA regulatory pathways: DDX6 mainly works through the miRNA pathway during early embryogenesis

Next, we aimed to identify which DDX6 pathway is most important during early development. DDX6 functions as a hub of post-transcriptional regulation. Due to its wide range of involvement, it is difficult to pinpoint which pathway is responsible when DDX6 is depleted. Therefore, we individually disrupted three main DDX6-associated pathways by knocking out a key gene of each pathway (S3C Fig). Translational repression along with P-body formation was impaired in *Eif4enif1* KO, 5'-to-3' mRNA degradation was impaired in *Dcp2* KO, and miRNA-mediated gene silencing was disrupted in *Dgcr8* KO [64–67,17].

To assess the effects on the transcriptomic landscape over the time course of development, we generated cDNA libraries of ESC and EpiLC Day2 stage from all mutant groups and WT.

Principle component analysis (PCA) demonstrated that *Ddx6* KO (*Ddx6*$^{\triangle/\triangle}$) is greatly different from WT, *Eif4enif1* KO, and *Dcp2* KO, but highly similar to *Dgcr8* KO (Fig 6A). *Eif4enif1* KO and *Dcp2* KO, whose protein functions directly affect P-body functions, resulted in similar transcriptomes. *Eif4enif1* KO, in which translational repression on transcripts is disrupted, resulted in the disassembly of P-bodies, whereas *Dcp2* KO caused the enlargement of P-bodies due to the blockage of mRNA degradation (S3D Fig). Even though their transcriptomes were different from WT, gastrulation occurred normally in KO embryos (S5A Fig). Based on the above, we concluded that P-bodies are dispensable until the peri-gastrulation stage. Disruption of P-body functions altered the transcriptome of cells, but the ultimate differentiation capacity of pluripotent stem cells was not affected. We examined differentially expressed genes in detail through gene set enrichment analysis (GSEA). A greater number of gene sets was affected in *Ddx6* KO and *Dgcr8* KO than in *Eif4enif1* KO and *Dcp2* KO (Fig 6B). The high portion of the differentially expressed gene sets of *Ddx6* KO ESCs was shared with *Dgcr8* KO ESCs (87% of upregulated, 84% of downregulated), indicating that DDX6 function is largely inclusive to the miRNA pathway. To identify the most affected gene sets, we filtered with adjusted p-value 0.05 and then aligned in order of normalized enrichment score (NES). The list of Top10 upregulated and downregulated gene sets in three different groups is shown in Table 2: *Ddx6* KO and *Dgcr8* KO only (regulated by miRNAs/ P-body-independent), all four KO groups (regulated by miRNAs/P-body-dependent), *Eif4enif1* KO and *Dcp2* KO only (regulated by P-body functions/miRNA-independent). We cannot go over these lists in detail, but they provide useful information for the functions of each pathway (Tables 2 and S1–S5).

As *Ddx6* KO and *Dgcr8* KO had similar transcriptomic changes in both ESC and EpiLC stages, we further examined their similarity in embryo samples. We generated *Dgcr8* KO embryos by injecting *Dgcr8* targeting sgRNAs into eggs via electroporation. *Dgcr8* KO embryos were smaller and more malformed than *Ddx6* KO embryos (Fig 6C), but we were able to find common defects with *Ddx6* KO: ectopically expanded and increased *Nodal* and *Eomes* expression in E7.5 embryos (Fig 6D). Unlike *Ddx6* KO, which exhibited a delay in primitive streak induction, *Dgcr8* KO embryos had a relatively normal and strong Brachyury induction at E7.5 despite their small body size. We also conducted detailed analyses during ESC-to-EpiLC differentiation through qRT-PCR. *Dgcr8* KO cells exhibited similar characteristics to *Ddx6* KO cells: enhanced pluripotency and stronger expression of the neural lineage-inducing factors with a decreased differentiation capacity to the mesendoderm lineage (Fig 6E). As we identified that repression of BMP signaling is the primary change in cellular condition of *Ddx6* KO, we examined *Dgcr8* KO for a similar change. *Dgcr8* KO cells also had upregulated expression of the negative regulators of BMP signaling, and pSMAD1/5 target genes were de-repressed. Moreover, GSEA of ESC RNA-seq data revealed that 'the negative regulation of BMP signaling' gene set (*Fbn1*, *Grem2*, *Grem1*, *Wnt5a*, *Htra1*, *Sorl1*, *Fstl3*, *Bmper*, *Nbl1*, *Htra3*, *Crim1*, *Rbpms2*, *Spart*) was highly upregulated only in *Ddx6* KO and *Dgcr8* KO ESCs, but not in *Eif4enif1* KO and *Dcp2* KO (Fig 6F). As such, DDX6-mediated RNA regulation and miRNA-mediated gene silencing share a common role, especially in preventing aberrant upregulation of the negative regulators of BMP signaling.

## Discussion

This study delineated the essential role of DDX6 in proper cell lineage specification and differentiation during early mouse embryogenesis (Fig 7A and 7B). We propose that DDX6 prevents cells from activating negative regulation of BMP signaling through miRNA-mediated gene silencing. The genes that are related to 'negative regulation of BMP signaling' were upregulated in both *Ddx6* KO E8.5 embryos and ESCs. Their phenotypes, including posterior body and

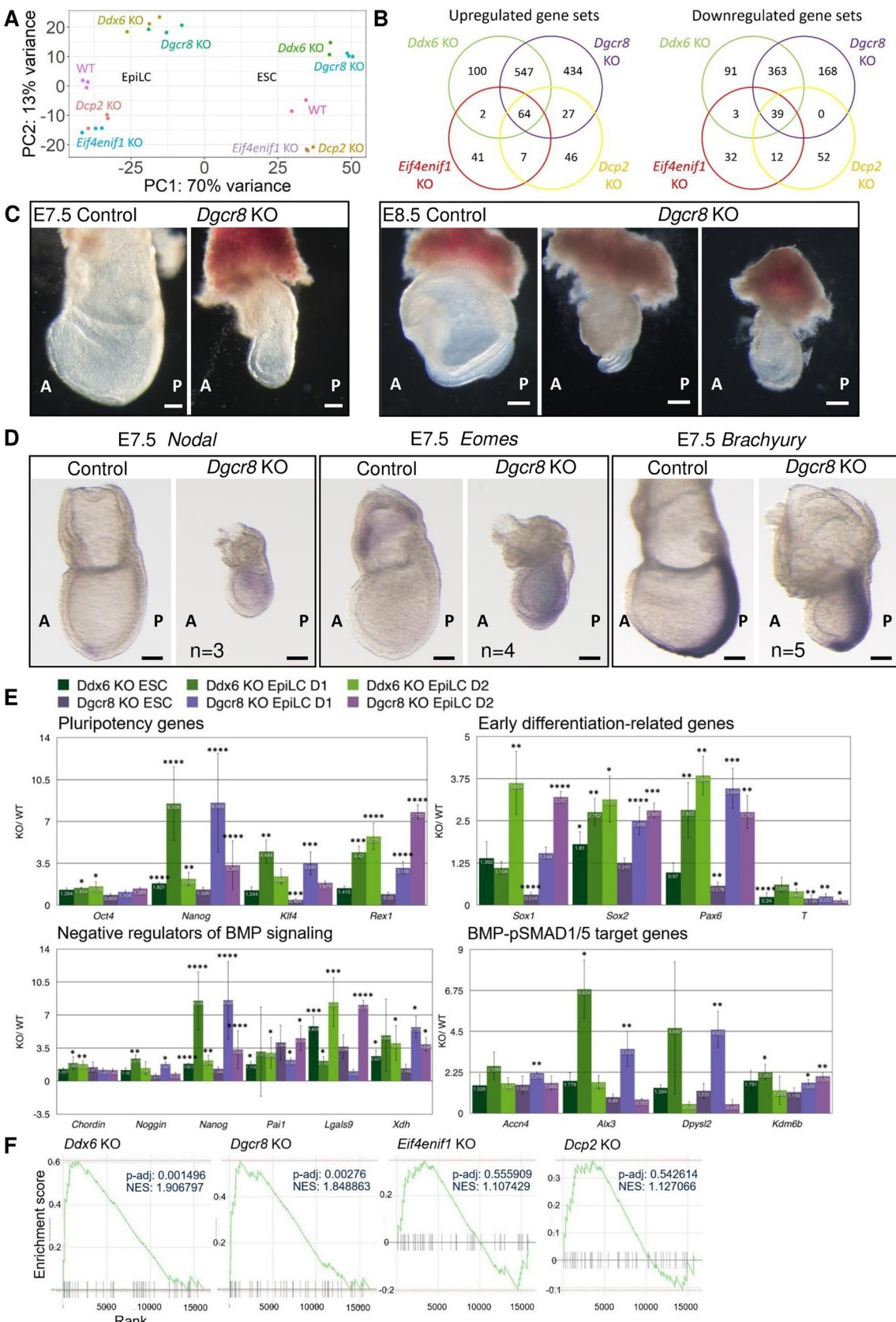

**Fig 6. Genetic dissection of the DDX6 functions indicates that the DDX6-miRNA pathway is essential during early embryogenesis.** (A) PCA plot of the ESC and EpiLC Day2 samples of each genotype group. (B) Summary of the GSEA results of ESC RNA-seq data. The gene sets that were changed in *Ddx6* KO, *Dgcr8* KO, *Eif4enif1* KO, and *Dcp2* KO were compared. (C) *Dgcr8* KO embryos exhibited similar morphological defects to *Ddx6* KO embryos at E7.5, but they became more malformed at E8.5 (Scale: 100 μm for E7.5; 200 μm for E8.5). (D) Whole-mount ISH of E7.5 *Dgcr8* KO embryos with *Nodal*, *Eomes*, and *Brachyury* probes (Scale: 100 μm, n = 3 for *Nodal*, n = 4 for *Eomes*, n = 5 for *Brachyury*). (E) Comparison of gene expression between *Ddx6* KO and *Dgcr8* KO. qRT-PCR analysis of several key genes during the EpiLC induction period. Each bar represents the relative expression of KO cells to WT cells at the indicated time point. Mean ± SEM. Student's t-test (n ≥ 3) (*p ≤ 0.05, **p ≤ 0.01, ***p ≤ 0.001, ****p ≤ 0.0001). (F) GSEA enrichment plot of the "negative regulation of BMP signaling pathway" gene set in four mutant ESCs. Black bars represent the position of the genes that belong to this gene set (n = 45) in the whole ranked gene list. The green line shows the overall distribution of this gene set (whether over-represented at the top (left) or bottom (right) of the ranked list of genes).

mesoderm developmental defects, premature neural induction, and de-repression of BMP-SMAD1/5 target genes, indicate that the BMP pathway is dysfunctional in *Ddx6* KO. Another characteristic of *Ddx6* KO was the increased expression of *Nodal* and its downstream targets. Promoted pluripotency, such as increased *Nanog* expression, and disruption of the primitive streak patterning by overexpanded *Eomes* expression can be attributable to high *Nodal*. However, there was no increase in the level of SMAD2 phosphorylation in *Ddx6* KO ESCs (S4A Fig). We now hypothesize that the increased expression of *Nodal* and its known downstream targets *Nanog* and *Eomes* in *Ddx6* KO is due to an altered SMAD3 activity level. Previously, Liu et al. reported a marked difference between SMAD2 and SMAD3. They showed that the main mediator of TGF-β signaling is SMAD2, whereas SMAD3 has a TGF-β signaling- and SMAD4-independent working mechanism [68]. Thus, we briefly analyzed the

**Table 2. Gene sets that are highly enriched in three different conditions.**

| Common in only *Ddx6* KO & *Dgcr8* KO ESCs (regulated by miRNAs/ P-body-independent) | Common in all four KO ESCs (regulated by miRNAs/ P-body-dependent) | Common in only *Eif4enif1* KO & *Dcp2* KO ESCs (regulated by P-body functions/ miRNA-independent) |
|---|---|---|
| Upregulated gene sets Top10 | | |
| 1.Serine type endopeptidase inhibitor activity | 1.Collagen-containing extracellular matrix | 1.Antioxidant activity |
| 2.Cytokine binding | 2.External encapsulating structure organization | 2.Response to toxic substance |
| 3.Positive regulation of chemokine production | 3.Extracellular matrix structural constituent | 3.Response to topologically incorrect protein |
| 4.Chemokine production | 4.Collagen fibril organization | 4.Cellular response to topologically incorrect protein |
| 5.Monocyte chemotaxis | 5.Collagen trimer | 5.Retinol metabolic process |
| 6.Negative regulation of cellular response to growth factor stimulus | 6.Lysosomal lumen | 6.Response to xenobiotic stimulus |
| 7.Proteoglycan metabolic process | 7.Collagen binding | 7.Pigment granule |
| 8.Immune receptor activity | 8.Basement membrane | |
| 9.Receptor complex | 9.Endoplasmic reticulum lumen | |
| 10.Regulation of leukocyte migration | 10.Vacuolar lumen | |
| Downregulated gene sets Top10 | | |
| 1.Ribosome | 1.Ribonucleoprotein complex biogenesis | 1.Regulatory T cell differentiation |
| 2.Ribosomal subunit | 2.Ribosome biogenesis | 2.Anterior posterior pattern specification |
| 3.Structural constituent of ribosome | 3.rRNA metabolic process | 3.Glandular epithelial cell development |
| 4.Mitochondrial gene expression | 4.ncRNA processing | 4.Microtubule polymerization |
| 5.Mitochondrial protein-containing matrix | 5.Ribonucleoprotein complex subunit organization | 5.Regulation of axonogenesis |
| 6.Translational termination | 6.ncRNA metabolic process | 6.Regionalization |
| 7.Catalytic step 2 spliceosome | 7.RNA splicing via transesterification reactions | 7.Positive regulation of axonogenesis |
| 8.Mitochondrial translation | 8.Methyltransferase complex | 8.Microtubule polymerization or depolymerization |
| 9.Translational elongation | 9.mRNA processing | 9.Regulation of microtubule cytoskeleton organization |
| 10.Mitochondrial translational termination | 10.Viral gene expression | 10.Regulation of DNA binding |

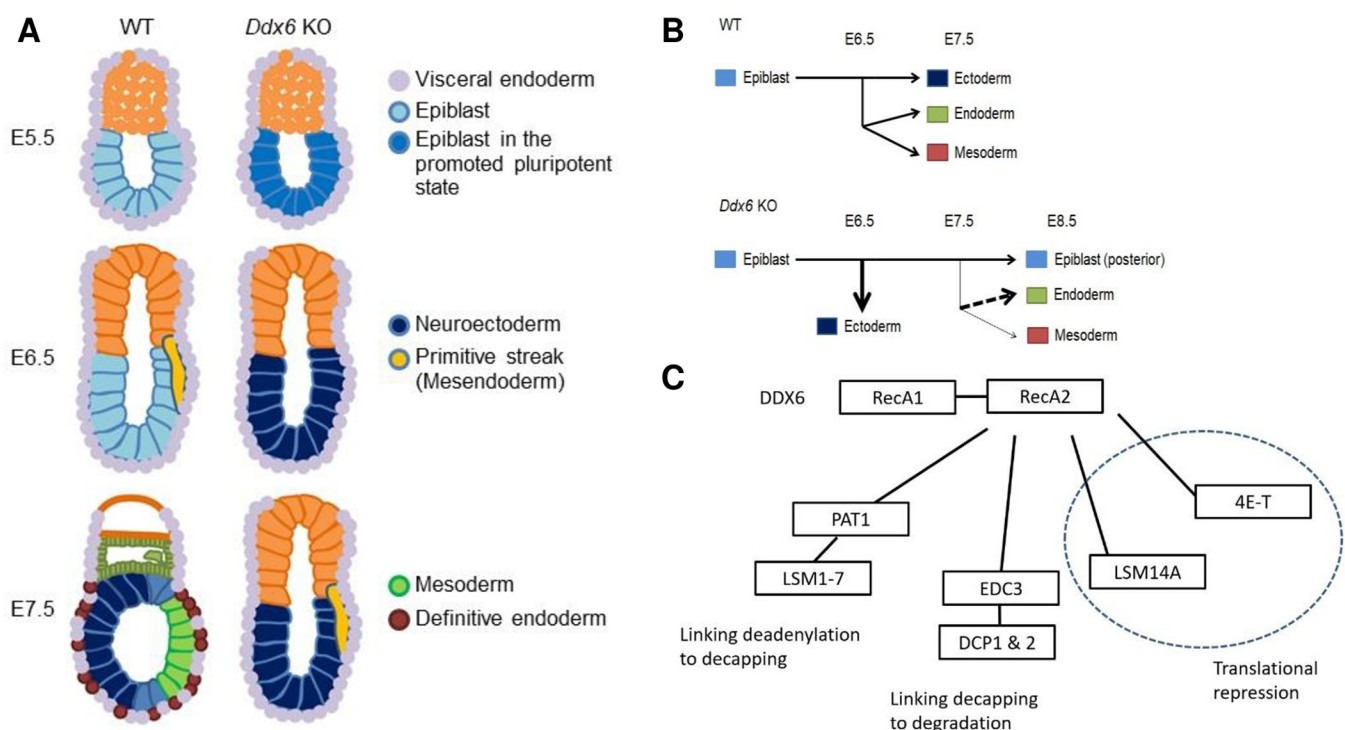

**Fig 7. Schemes showing developmental defects caused by loss of DDX6-mediated RNA regulation.** (A) Development of the three primary embryonic germ layers is largely affected by DDX6. Neuroectoderm is specified earlier than WT, whereas formation of the primitive streak is delayed (The smaller size of the *Ddx6* mutant is not reflected in the images). (B) Changes in cell-lineage specification from pluripotent stem cells caused by *Ddx6* loss are depicted on a horizontal diagram. Uncommitted *Ddx6*$^{\triangle/\triangle}$ pluripotent cells possess promoted pluripotency and strongly favor commitment to the neuronal lineage. In WT embryos, the mesendoderm lineage arises at ~E6.5 as the primitive streak is formed, and three germ layers simultaneously develop at ~E7.5. In *Ddx6* KO embryos, premature neural induction occurs with a one-day delay in primitive streak formation. During mesendoderm segregation, definitive endoderm specification is increased, whereas mesoderm specification is greatly reduced due to the patterning defect of the primitive streak. Posterior epiblast cells cannot exit pluripotency on time, impeding the differentiation processes. (C) The interaction of DDX6 with other gene silencing effector protein complexes.

public ChIP-seq data for SMAD2 and SMAD3 in embryoid bodies, which are similar to early-stage embryos. When we assayed potential target genes using 'ChIP-Atlas,' *Eomes*, whose expression was significantly increased in *Ddx6* KO embryos, was only bound by SMAD3 (S4B and S4C Fig). *Nanog* was more preferentially bound by SMAD3, whereas *Nodal* was equally regulated by SMAD3 and SMAD2. Based on these results, *Ddx6* KO phenotypes may arise from the increased SMAD3-induced transcriptional activation rather than Nodal signaling itself. Further detailed analyses are required to clarify this possibility.

Since DDX6 functions in multiple pathways, it is difficult to find the most crucial one for a certain cellular process. To overcome this problem, we examined three gene-regulating pathways, in which DDX6 is known to have a role: translational repression, mRNA degradation, and miRNA-mediated gene silencing by knocking out the key gene.

EIF4ENIF1 (4E-T, eIF4E-Transporter) is a kind of eIF4E-binding proteins (4E-BPs), which inhibit the initiation step of translation by binding to eIF4E, a component of the translational initiation factor eIF4F complex [69–72]. When eIF4E is captured by 4E-T, an mRNA whose cap is bound by eIF4E is also brought into P-bodies. Importantly, eIF4E-4E-T-bound transcripts are protected from decapping even after deadenylation. Hence it is a distinct mechanism that specifies mRNAs to be stored in P-bodies instead of being degraded [73]. 4E-T binds to DDX6 via Cup homology domain (CHD) and is necessary for the assembly of P-bodies [17]. Thus, we wanted to examine the degree of involvement of DDX6 in translational repression process and the dependency of DDX6 functions on P-body formation via *Eif4enif1* KO.

DCP2 (NUDT20) is a major mRNA decapping enzyme in mammalian cells along with another Nudix hydrolase protein, NUDT16 [74–76]. Knockdown of *DCP2* in human U2OS cells caused the enlargement of P-bodies with the accumulation of the MS2-tagged β-actin transcripts, leading to the conclusion that a constant mRNA decay occurs inside the P-bodies, and when DCP2 is depleted, this decay process is inhibited [67]. DCP2 is connected to DDX6 through EDC3 [77]. As *Dcp2* KO also changes the morphology of P-bodies, we were able to study the effects of the disruption of mRNA decay pathway as well as functional defect of P-bodies through *Dcp2* KO.

Even though this genetic approach cannot directly explore molecular functions of P-bodies or target transcripts, we tried to indirectly examine their roles in cellular processes by conducting transcriptomic analyses. The RNA-seq analyses identified the similarity between *Ddx6* KO and *Dgcr8* KO. Only *Dgcr8* KO, but neither *Eif4enif1* KO nor *Dcp2* KO, closely phenocopied *Ddx6* KO, indicating that DDX6 mainly works through the miRNA pathway among the various RNA regulatory means during early embryogenesis. The double-stranded RNA-binding protein DGCR8 forms the microprocessor complex with the RNase III enzyme Drosha to process long primary miRNAs (pri-miRNAs) into precursor miRNAs (pre-miRNAs) [78,79]. In *Dgcr8* KO, miRNA-mediated gene silencing becomes nonfunctional because of the failure of miRNA generation [66]. miRNAs repress target gene expression through either translational repression or mRNA degradation [80–83]. In mESCs, the loss of DDX6 impaired only miRNA-induced translational repression [48]. Thus, the greater number of upregulated gene sets in *Dgcr8* KO ESC than in *Ddx6* KO (1211 vs 828 gene sets) may have resulted from the disruption of miRNA-mediated mRNA destabilization. DDX6 participates in the effector step of miRNA-mediated gene silencing by binding to CNOT1, the scaffold subunit of the CCR4-NOT complex. The CCR4-NOT complex, a predominant generic deadenylase [84], cooperates with miRISC by binding to GW182/TNRC6 protein [80], and then recruits DDX6. The binding of CNOT1 and DDX6 activates the ATPase activity of DDX6, and this change is significant for target gene repression [85]. Our study is connected to previous studies which demonstrated the important role of miRNAs during early embryogenesis. miR-302 and miR-290 are highly expressed in pluripotent cells and are required for neurulation. Loss of miR-302 causes precocious neuronal differentiation [86]. Moreover, miR-302 stimulates BMP signaling by repressing BMP inhibitors TOB2, DAZAP2, and SLAIN1 in hESCs [87]. Although these genes were not upregulated in *Ddx6* KO or *Dgcr8* KO mESCs, probably due to the species difference, it is a good example that miRNAs can indirectly affect BMP signaling by targeting BMP signaling inhibitors.

Embryos had abundant P-bodies from blastocyst to peri-gastrula embryos, but disruption of P-body function had marginal effects on early embryogenesis. Based on our results, we refute Di Stefano et al. [20]'s statement that disruption of P-bodies is the cause of *Ddx6*-deficient PSCs being "hyper-pluripotent." Ayache et al. [17] investigated multiple conditions for P-body assembly in HEK293 cells and found that DDX6, 4E-T (EIF4ENIF1), and LSM14A (RAP55) are the three key factors of P-body formation. *Eif4enif1* KO mESCs from our study and *LSM14A* knockdown hESCs from Di Stefano et al. completed the puzzle of the relationship between DDX6 and the P-body. The characteristics of *Ddx6* KO mESCs and *DDX6* KD hESCs were similar. As described earlier, our mESCs also exhibited hyper-pluripotent properties by having high expression of *Nanog* and failing to exit the ground state properly. However, the two P-body mutants exhibited different phenotypes. Knockdown of *LSM14A* made hESCs hyper-pluripotent with the significantly higher expression of *NANOG* like *DDX6*-depleted hESCs. In our case, deletion of *Eif4enif1* led to the disassembly of P-bodies in mESCs (S3D Fig), but *Eif4enif1* KO pluripotent cells did not display hyper-pluripotent properties (S5B Fig). Furthermore, *Eif4enif1* KO embryos lacked the developmental defects of *Ddx6* mutants (S5A

Fig). As such, P-bodies are not essential for development at least until the gastrulation stage. This result is consistent with a previous study demonstrating that P-body formation is not the cause but a consequence of RNA-mediated gene silencing activities [13]. However, we do not totally agree that P-bodies are merely a resultant complex. Our RNA-seq analyses showed that the gene expression profiles of *Eif4enif1* KO and *Dcp2* KO moved into a similar direction and clustered together despite the apparent functional differences. We speculated that this change may have come from the disruption of overall P-body function. From the GSEA, there were a few gene sets that are commonly changed only in *Eif4enif1* KO and *Dcp2* KO. As these gene sets were not affected in *Ddx6* KO or *Dgcr8* KO, we considered them as genes that are associated with P-bodies but not related to miRNA-mediated gene silencing (Tables 2 and S1). miRNA-mediated gene silencing activity was functional when P-bodies were disrupted, suggesting miRNA function is P-body-independent [13]. However, the presence of distinctive gene sets that are altered by both miRNA pathways (*Ddx6* KO & *Dgcr8* KO) and P-body-related functions (*Eif4enif1* KO & *Dcp2* KO) may suggest that GW182 and AGO2, the essential components of the miRISC, locate to P-bodies for certain functional purposes rather than just being stored there.

Unlike in plants, direct cleavage by AGO2 endonuclease of miRISC is rare in animal cells [88,89]. Animal miRISCs instead recruit other effector proteins to facilitate either translational repression or mRNA decay [90]. We conjectured that DDX6, which interacts with various mRNA repressing proteins and is incorporated into the miRISC, would work as a bridge coupling miRNA target with the RNA repression effector. We initially expected that depletion of 4E-T (disruption of translational repression pathway) or DCP2 (disruption of mRNA decay pathway) would show the similar phenotypes to DDX6 loss. Different from our expectation, these downstream effector pathways did not phenocopy *Ddx6* KO and *Dgcr8* KO. On the other hand, the loss of LSM14A resulted in a similar phenotype to *Ddx6*-depleted cells [20]. These results led to the following hypotheses: 1) According to GSEA, distinct gene sets were commonly affected in *Eif4enif1* and/or *Dcp2* KO (Tables 2 and S1). These gene sets may not be important for early developmental processes. 2) There are diverse translational repressors in the cytoplasm. The 4E-T-associated mechanism may not be the major method of translational repression in pluripotent cells. DDX6 uses the same surface (RecA2) for binding to different RNA regulatory proteins in a mutually exclusive manner (Fig 7C) [91–97,8,77,21,11]. Among various binding partners, 4E-T and LSM14A are the two translation repression proteins whose interaction with DDX6 is well documented [21,11,97]. Taken together, we surmise that the DDX6-LSM14A complex is necessary for facilitating translation repression targeted by miR-NAs, but P-body assembly/maintenance is not required for this process.

Although DDX6, miRNA-mediated gene silencing (DGCR8), 4E-T, DCP2 are highly inter-correlated, they also have other molecular functions independent of P-bodies or proteins that are analyzed here. To have a clue about each protein's unique role, we searched genes that were only affected in each KO ESCs (Fig 6B and S2–S5 Tables). For instance, in *Ddx6* KO ESCs (S2 Table), some of upregulated gene sets were related to the immune response (MYD88-independent toll like receptor signaling pathway, positive regulation of small GTPase-mediated signal transduction, myeloid dendritic cell activation, positive regulation of leukocyte apoptotic process), which is consistent with the known DDX6 role of preventing aberrant activation of interferon-stimulated genes [98]. Many downregulated gene sets were related to nucleoside and nucleotide biosynthesis and metabolism, which implies that DDX6 is deeply involved in nucleoside/nucleotide processing.

Lastly, we discuss the significance of DDX6 studies. DDX6 is ubiquitously expressed with well conserved functional capabilities, but its function often becomes specific in a certain cell type depending on interacting protein complexes. For example, DDX6 is necessary for

parental RNA degradation in fibroblasts for iPSC reprogramming in concert with the RO60/ *RNY1* complex [99]. It also acts as a promoter of translation in epidermal progenitor cells along with the YBX1/EIF4E complex [15]. DDX6 has another role in packaging the retroviral RNA genome by interacting with the viral RNA-Gag ribonucleoprotein complex [100]. One of the key findings of our study is that DDX6 molecular functional pathways can be relatively clearly segregated. We need to exploit this potent endogenous gene expression regulator. DDX6 participates in various RNA metabolisms and has a wide expression pattern (e.g., in pluripotent cells, adult tissue stem/progenitor cells, cancer, and RNA-virus infected cells). Deciphering DDX6-mediated regulatory pathways and modular manipulation of a specific pathway may make DDX6 a useful genetic therapeutic target.

## Materials and methods

### Ethics statement

All mouse experiments (R2-4 and R3-8) were approved by the National Institute of Genetics (NIG) Institutional Animal Care and Use committee (Tsuyoshi Koide, Koichi Kawakami, Yumiko Saga, Tatsumi Hirata, Shoko Kawamoto, Naoki Nakagawa, Susumu Sano, Hiroyasu Furuumi, and Nobuyoshi Shiojiri).

### Mice

Mice were housed in a specific-pathogen-free animal care facility at the NIG. $Ddx6^{\triangle/+}$, $Ddx6^{flox/flox}$, $Rosa\text{-}CreER^{T2}$, $Ddx6\text{-}mCherry$, $Eif4enif1$ KO, $Dgcr8$ KO, *and* $Dcp2$ KO mice were used in this study. The production strategy for the $Ddx6^{flox/flox}$ mouse line is described in [47]. The $Ddx6^{\triangle/+}$ mouse line was generated from the $Ddx6^{flox/flox}$ mouse line by removing floxed exon5 through crossing with the deletion mouse line CAG-Cre [101]. Genotyping was carried out using the primers [Ddx6-LA-Fw1: TTGTGCTGGGATGAGCCTAC; Ddx6-RA-Rv1: AGTTGCATCAACGACAGGAGAG]. The $Ddx6\text{-}mCherry$ reporter mouse was established at NIG by injecting a targeting vector containing mCherry with homology arms of the $Ddx6$ gene with Cas9-gRNA designed at the C-terminal of $Ddx6$ gene and Cas9 protein. Genotyping was carried out using the primers [mCherry-L1: GGAACAGTACGAACGCGCCG; DDX6-GR1: GACAGGTGCATGTGTTCACCC]. $Eif4enif1$ KO mice and $Dgcr8$ KO mice were directly obtained as F0 generation, which were produced by delivering Cas9 protein and guide RNAs targeting ("TCTGGTTCATACCGTAGTTT", "AACTTACTTTCGTATAGCGA" for Eif4enif1 exon2) and ("TGAATCCTAATTGCACCCGT", "GAACAGGAAGCATACGGG TA", "TGGGTCGGTCTGCAGAGTTG" for $Dgcr8$ exon4 & 5) into the fertilized eggs via electroporation. A similar strategy was used to establish the $Dcp2$ KO mouse line; two gRNAs targeting "AACAAAGCCAACCCGG" and "CGCGGCACTGAAGTGT" and Cas9 protein were delivered via electroporation. The mouse line with a correctly deleted exon2 was selected and expanded. The homozygous KO pups were acquired by crossing $Dcp2$ heterozygous mice.

　• For conditional deletion of the floxed $Ddx6$ alleles, 600 µL of 10 mg/mL tamoxifen was administered to pregnant females.

### Establishment of ES cells

$Ddx6$ KO: $Ddx6^{\triangle/+}$ mice were intercrossed and blastocysts were collected from the uterus on E3.5. Collected blastocysts were cultured on mitomycin-treated mouse embryonic fibroblast feeder cells in 2i-LIF medium (ESGRO Complete Basal medium (Millipore, Germany) supplemented with leukemia inhibitory factor (Wako, Tokyo, Japan), 0.4 µM MEK inhibitor PD0325901 (Wako, Tokyo, Japan), 3 µM GSK3 inhibitor CHIR99021 (Wako, Tokyo, Japan),

and Penicillin-Streptomycin (Invitrogen). Blastocyst outgrowths were disaggregated and passaged onto the new wells plated with feeder cells in the same medium condition. Once ES cell colonies developed, they were expanded for genotyping and storage. Genotyping was carried out using the primers [Ddx6-LA-Fw1: TTGTGCTGGGATGAGCCTAC; Ddx6-RA-Rv1: AGTTGCATCAACGACAGGAGAG].

*Dcp2* KO: First, the *Dcp2* cKO ES line was established by replacing the endogenous exon2 with the floxed exon2 through CRISPR/Cas9-mediated homologous recombination. The *Dcp2* KO ES line was acquired by incubating cKO ESCs in culture medium containing 4-hydroxytamoxifen (4-OHT). Genotyping was carried out using the primers [Dcp2-LA-Fw1: TTCTGCTGCTTTCAAGCCTGG; Dcp2-int2-R2: ACATTCGCTACAACAACGCTTC].

*Eif4enif1* KO was also generated by deleting floxed exon2 from conditional KO ESCs and replacing the endogenous exon2 with the floxed exon2 through CRISPR/Cas9-mediated homologous recombination. Genotyping was carried out using the primers [4ET-int1-F1: GTGACAGGCACTTTCCAGCAG; 4ET-int2-R1: TTCCAAAGCCTTAGCTGCTTCTC].

*Dgcr8* KO was established by deleting exon4 and a part of exon5 using two Cas9 vectors targeting "TGAATCCTAATTGCACCCGT" and "TGGGTCGGTCTGCAGAGTTG." CRISPR direct (http://crispr.dbcls.jp/)[102] was used to find Cas9 target sites, and the target sequence was integrated into a modified px330 Cas9 vector (Addgene), which contains the pgk-puromycin cassette. ESC transfection was performed with Lipofectamine 2000 (Invitrogen). Genotyping was carried out using the primers [Dgcr8-int3-F2: GCTCCTGGAGTAGGCATGTTG; Dgcr8-ex5-R1: TTCACTTGTCCCAGGGCTCC].

We confirmed successful targeting by checking the target gene loci using the Integrative Genomics Viewer (IGV)[103] (S5C Fig).

## Immunostaining

For frozen section immunohistochemistry, embryos were fixed in 4% paraformaldehyde for 30 min at 4˚C, submerged in 10% sucrose for 1~2 hours, in 20% sucrose overnight at 4˚C, and frozen in Tissue-Tek O.C.T. compound (Sakura Finetek, Tokyo, Japan). Each 6-μm-thick section was applied to glass slides. After blocking with 3% skim milk in PBS-T (PBS with 0.1% Tween20 (Sigma-Aldrich)) at room temperature for 45 min, samples were incubated with primary antibodies overnight at 4˚C. The primary antibodies are listed in Table 3. The next day, samples were washed with PBS-T and incubated with secondary antibodies labeled with Alexa

**Table 3. Antibodies.**

| Antigen | Usage | Concentration | Manufacturer | Reference # |
|---|---|---|---|---|
| Rck/p54 (DDX6) | IHC, ICC | 1:300 | MBL | PD009 |
| DCP1A | IHC, ICC | 1:200 | ABNOVA | H00055802-M06 |
| BRACHYURY | IHC | 1:400 | R&D SYSTEMS | O15178 |
| NANOG | IHC | 1:200 | NOVUS BIOLOGICALS | NB100-58842 |
| TuJ1 | IHC | 1:1500 | Abcam | Ab18207 |
| SOX1 | IHC | 1:100 | NOVUS BIOLOGICALS | AF3369 |
| SOX2 (Y-17) | IHC | 1:200 | SANTA CRUZ BIOTECHNOLOGY | sc-17320 |
| DCX | IHC | 1:400 | Cell Signaling TECHNOLOGY | (A8L1U) 14802 |
| Phospho-Smad1/5 (Ser463/465) | Western | 1:1000 | Cell Signaling TECHNOLOGY | 9516 |
| Smad5 | Western | 1:2000 | abcam | Ab40771 |
| Phospho-smad2 (Ser465/467) | Western | 1:2000 | Cell Signaling TECHNOLOGY | 3101 |
| smad2/3 | Western | 1:2000 | Cell Signaling TECHNOLOGY | 8685 |
| β-actin | Western | 1:5000 | Sigma-Aldrich | A5316 |

Fluor 488, 594, or 647 (1:1000 dilution in PBS-T, Invitrogen) for 1 hr 10 min at room temperature. DNA was counterstained with DAPI (100 ng/mL). Images were acquired by the Olympus FV1200 confocal microscope and processed with FV10-ASW (version 4.0) software.

For whole-mount immunostaining, embryos were fixed in 4% paraformaldehyde for 30 min at 4˚C, and permeabilized with 1% Triton X-100 in PBS for 30 min. Blocking was done with 10% FBS and 1% BSA for 1 hr at room temperature. Samples were then incubated with primary antibodies overnight at 4˚C, and the secondary antibody reaction was performed overnight at 4˚C. Images were taken by the Olympus FV1200 confocal microscope.

For immunocytochemistry, cells were fixed in 4% paraformaldehyde for 12 min at room temperature, permeabilized with 0.3% Triton X-100 in PBS for 12 min at room temperature, and blocked with 3% skim milk in PBS-T for 45 min at room temperature. After blocking, cells were incubated with primary antibodies overnight at 4˚C. The following day, samples were incubated with secondary antibodies for 1 hr at room temperature and counterstained with DAPI. Images were acquired by the Olympus FV1200 confocal microscope or Leica DM6000 FS light microscope.

## Western blotting

$3 \times 10^5$ WT and *Ddx6* KO ESCs were plated into the well of a 12-well plate with feeder cells in 2i-LIF medium. Two days later, cells were treated with 10 ng/ml rBMP4, 0.6 µM LDN193189, or 10 mM SB431542 (FUJIFILM/Wako) for 3 hrs, and 25 ng/ml of ActivinA for 1 or 2 hrs at 37˚C. ESCs were harvested with TNE buffer (50 mM Tris-HCl pH 7.4, 150 mM NaCl, 1 mM DTT, 1 mM EDTA, and 1% NP40) supplemented with cOmplete Protease inhibitor cocktail (Roche), and centrifuged. Next, the supernatant concentration was adjusted, mixed with 3XSDS buffer (0.2M Tris-HCl pH6.8, 9% SDS, 30% glycerol, 15% 2-mercaptoethanol, 0.006% bromophenol blue), and incubated at 96˚C for 5 min. After SDS-PAGE, proteins were transferred onto a Immobilon-P Transfer Membrane (Millipore). The membrane was blocked with 3% skim milk /TBS-T, and then incubated with primary antibodies diluted in Can Get Signal 1 (TOYOBO, NKB101) for phospho-Smad1/5 or phospho-Smad2 antibodies or 3% skim milk /TBS-T for other antibodies at 4˚C overnight and subsequently incubated with a secondary antibody at room temperature for 90 min. Detection was performed using the SuperSignal West Femto Maximum Sensitivity Substrate (Thermo Scientific) and images were acquired with a ChemiDoc Touch MP system (Bio-Rad).

## Whole-mount *in situ* hybridization

Probe generation and whole-mount embryo ISH procedures were performed according to a previously reported protocol [104].

## RNA-seq

**E8.5 embryos.**   RNA was collected from E8.5 embryos using TRIzol Reagent (Thermo Fisher Scientific). Genotyping was done with extraembryonic tissue or yolk sac and further confirmed by qPCR after acquiring cDNA. 300 ~ 420 ng of RNA was used for E8.5 cDNA library generation with the KAPA-Stranded mRNA-seq kit (Illumina Platforms, KR0960-v5.17).

## ESC & EpiLCs

RNA of WT, *Ddx6* KO, *Eif4enif1* KO, *Dcp2* KO, and *Dgcr8* KO ESC and EpiLC Day2 samples was extracted using RNAiso Plus (Takara, Tokyo, Japan) according to the manufacturer's

instructions. RNAs were selected by polyA. Two ESC cDNA libraries (three *Dgcr8* KO ESC) & three EpiLC Day2 cDNA libraries were generated for each genotype group. cDNA libraries were generated using the TruSeq Stranded mRNA kit (illumina, 20020595) following the accompanying protocol. Samples were sequenced on a NovaSeq 6000 (101 bp paired-end sequencing), averaging 47 million read pairs per sample.

## Bioinformatics analysis

**E8.5 embryos.** For all libraries, low-quality sequences, adapters, and polyA or T were trimmed or removed using Cutadapt (version 2.8) [105] with the following options: "-e 0.1 -q 20 -m 20 -O 3 -a GATCGGAAGAGCACACGTCTGAACTCCAGTCAC -a A{100} -a T{100}". The raw reads and processed reads were checked using FastQC (version 0.11.7, http://www. bioinformatics.babraham.ac.uk/projects/fastqc/). To map the reads to the mouse reference genome, the UCSC mm10 mouse reference genome (fasta) and gene annotation (General Transfer Format (GTF)) file were downloaded from Illumina iGenomes (https://sapac. support.illumina.com/sequencing/sequencing_software/iGenome.html). To increase the mapping accuracy of splicing reads, splicing-site and exon information were extracted from the gene annotation GTF file using the Python scripts hisat2_extract_splice_sites.py and hisat-t2_extract_exons.py, respectively, from the HISAT2 (version 2.1.0) package [106]. The HISAT2 index files of the reference genome were built including the extracted genomic information using "hisat2-build" command with options: "—ss" and "—exon". Clean reads were then mapped to the HISAT2 index files using the HISAT2 with default options. The obtained Sequence Alignment Map (SAM) files were sorted by genomic coordinates and converted to Binary Alignment Map (BAM) files using SAMtools (version 1.9) [107] "sort" command with option: "-O BAM". Raw read counts per gene were calculated using featureCounts (version 2.0.0) [108] with options: "-s 2 -t exon -g gene_id -a iGenomes/mm10/Annotation/Genes/ genes.gtf". Normalized counts were calculated by the trimmed mean of M-values (TMM) method using the Bioconductor edgeR (version 3.28.1) [109] in R (version 3.6.3) [https://cran. r-project.org/]. Principal component analysis (PCA) was performed on the $\log_2$ transformed normalized counts obtained from the "cpm" function in edgeR using the "prcomp" function with default options in R. Differentially expressed genes (DEGs) were detected using edgeR with the cut-off criteria of $\log_2$ (fold change) $> 1$ or $< -1$. Gene ontology term enrichment analysis was performed via Metascape [110].

## ESC & EpiLC of WT, *Ddx6* KO, *Eif4enif1* KO, *Dcp2* KO, and *Dgcr8* KO

For all libraries, low-quality sequences and adapters were trimmed or removed using fastp (version 0.20.0) [111] with the following options: "-G -3 -n 1 -l 80". For preparation for mapping reads to the mouse reference genome, the Ensembl mouse reference genome (release-102, Mus_musculus.GRCm38.dna_sm.primary_assembly.fa.gz) and the gene annotation GTF file (Mus_musculus.GRCm38.102.gtf.gz) were downloaded from the Ensembl ftp site (http:// ftp.ensembl.org/). To increase the mapping accuracy of splicing reads, splicing-site and exon information were extracted from the gene annotation GTF file using the Python scripts hisat-t2_extract_splice_sites.py and hisat2_extract_exons.py, respectively, from the HISAT2 (version 2.2.1) package. The HISAT2 index files of the reference genome were built including the extracted genomic information using "hisat2-build" command with options: "—ss" and "—exon". Clean reads were then mapped to the HISAT2 index files using the HISAT2 with default options. The obtained SAM files were sorted by genomic coordinates and converted to BAM files using SAMtools (version 1.13) "sort" command with option: "-O BAM". Raw read counts per gene were calculated using featureCounts (version 2.0.3) with options: "-s 2 -p—

countReadPairs -B -t exon -g gene_id -a Mus_musculus.GRCm38.102.gtf", and then low-abundance genes were removed by removing the genes with total number of mapped reads < 10 among 26 samples. Normalized counts and statistical values for differential gene expression analysis were calculated by the default settings through the steps: 1. estimation of size factors, 2. estimation of dispersion, and 3. Negative Binomial GLM fitting and Wald statistics using the Bioconductor DESeq2 packages (version 1.32.0) [112] in the R (version 4.1.1). To assess gene expression correlation between samples, the pair-wise scatter plot was produced using $\log_2$(the normalized counts + 1), the "cor" function with the parameter "method = 'spearman', use = 'pairwise.complete.obs'", and ggplot2 packages [113] in R. For PCA, the variance stabilizing transformed (vst) normalized counts were calculated using the vst function of DESeq2 with the default settings and PCA was performed with the top 500 most variable genes using the DESeq2 plotPCA function. To assess DEGs, MA plots for each comparison between sample groups were produced using the results of the DESeq2 analysis and the ggplot2. DEGs were detected using DESeq2 with the cut-off criteria of adjusted p-value < 0.05 and $\log_2$(fold change) > 2 or < -2. Gene ontology enrichment term analysis was performed via Metascape.

### Gene set enrichment analysis

Gene set enrichment analysis was implemented via the R package 'fgsea' [114,115] using the genes pre-ranked based on Wald statistic values of ESCs obtained from DESeq2 as 'stat'. The used gene set was the 'Biological Processes' gene set collection from MSigDB v7.4 [116]. To generate highly differentially expressed gene set tables (Tables 2 and S1–S5), the cutoff was made with the absolute NES value ≥ 1.5 and p-adj ≤ 0.05, and then the gene setlist was aligned in order of highest absolute NES in each direction.

### Public ChIP-seq data analyses (ChIP-Atlas database)

The potential targets of SMAD2 and SMAD3 in embryoid bodies were examined using the 'ChIP-Atlas: Target genes' function [117]: ChIP-Atlas. https://chip-atlas.org). The used data set name: [SMAD3] SRX5251393, SRX5251394 (n = 2) [SMAD2] SRX1080389, SRX1080390, SRX1080403, SRX1080404, SRX1080406, SRX1080407, SRX5251391, SRX5251392 (n = 8). The cutoff was made with the binding score 750.

### qRT-PCR

Embryos were frozen in RNAiso Plus (Takara) and extracted according to the manufacturer's instructions. RNA of cultured cells was extracted by RNeasy Mini Kits (Qiagen, Germany). Extracted RNA was treated with Recombinant DNaseI (Thermo Scientific) for 30 min at 37°C, and processed for reverse transcription using SuperScript III or IV Reverse Transcriptase (Invitrogen). Quantitative PCR was performed using KAPA SYBR Fast qPCR Kits (Nippon Genetics, Japan) on a Dice Real-Time System Single Thermal Cycler (Takara) or CFX96 Real-Time System (BioRad) machine. The primer sequences are listed in Table 4. The expression level was normalized to *Gapdh* and the relative expression was calculated by the $\Delta\Delta C_T$ method.

### Statistical analysis

Significance for *in vitro* experiments was examined by the Student's t-test. $^*p \leq 0.05$, $^{**}p \leq 0.01$, $^{***}p \leq 0.001$ and $^{****}p \leq 0.0001$. Significance for embryo RT-qPCR experiments was assayed by the Wilcoxon rank-sum test. $^*\alpha = 0.05$, $^{**}\alpha = 0.01$. Error bars represent s.e.m.

**Table 4. qPCR primer list.**

| Gene | Forward (5'-to-3') | Reverse (5'-to-3') |
|------|--------------------|--------------------|
| *Accn4* | AGGAGGCAGGGGATGAACA | TGAGGTGAGTAGGGCCAGTG |
| *Alx3* | GCTACCAGTGGATTGCCGAG | GCTCCCGAGCATACACGTC |
| *Cer1* | CTACAGGAGGAAGCCAAGAGGTTC | TGGGCAATGGTCTGGTTGAAGG |
| *Chordin* | CTGCGCTCAAGTTTACGCTTC | AGGGTGTTCAAACAGGATGTTG |
| *Dcx* | ATGCAGTTGTCCCTCCATTC | ATGCCACCAAGTTGTCATCA |
| *Ddx6* | TCCTATCCAGGAGGAGAGCATT | ATGAGGTAGGCACCGCTTTT |
| *Dpysl2* | CAGAATGGTGATTCCCGGAGG | CAGCCAATAGGCTCGTCC |
| *Eomes* | CCTTCACCTTCTCAGAGACACAGTT | TCGATCTTTAGCTGGGTGATATCC |
| *Fgf5* | GCTGTGTCTCAGGGGATTGT | CACTCTCGGCCTGTCTTTTC |
| *Fzd4* | TGCCAGAACCTCGGCTACA | ATGAGCGGCGTGAAAGTTGT |
| *Gapdh* | TGTGTCCGTCGTGGATCTGA | TTGCTGTTGAAGTCGCAGGAG |
| *Hes7* | ACCAGGGACCAGAACCTCC | GGCTTCGCTCCCTCAAGTAG |
| *Isl1* | AGATTATATCAGGTTGTACGGGATCA | ACACAGCGGAAACACTCGAT |
| *Kdm6b* | CCCCCATTTCAGCTGACTAA | CTGGACCAAGGGGTGTGTT |
| *Klf4* | CTTCAGCTATCCGATCCGGG | GAGGGGCTCACGTCATTGAT |
| *Lgals9* | GGCGCAAACAGAAAACTCAGAA | ACGGGTAAAGCCCATTTGGA |
| *Nanog* | TTGCTTACAAGGGTCTGCTACT | ACTGGTAGAAGAATCAGGGCT |
| *Nodal* | CCTGGAGCGCATTTGGATG | ACTTTTCTGCTCGACTGGACA |
| *Noggin* | GCCGAGCGAGATCAAAGG | TCTTGCTCAGGCGCTGTTT |
| *Oct4* | TCACCTTGGGGTACACCCAG | CATGTTCTTAAGGCTGAGCTGC |
| *Pai1* | TCATCAATGACTGGGTGGAA | TGCTGGCCTCTAAGAAAGGA |
| *Pax6* | GATAACATACCAAGCGTGTCATCAATA | TGCGCCCATCTGTTGCT |
| *Rex1* | ACGAGGTGAGTTTTCCGAAC | CCTCTGTCTTCTCTTGCTTC |
| *Sox1* | GCAGCGTTTCCGTGACTTTAT | GGCAGAACCACAGGAAAGAAA |
| *Sox2* | GCGGAGTGGAAACTTTTGTCC | CGGGAAGCGTGTACTTATCCTT |
| *Sox17* | GAGGGCCAGAAGCAGTGTTA | AGTGATTGTGGGGAGCAAGT |
| *T* | CTCGGATTCACATCGTGAGAG | AAGGCTTTAGCAAATGGGTTGTA |
| *Tbx6* | ATGTACCATCCACGAGAGTTGT | CCAAATCAGGGTAGCGGTAAC |
| *Xdh* | CGATGACGAGGACAACGGT | TGAAGGCGGTCATACTTGGAG |
| *Zic3* | CAAGAGGACCCATACAGGTGAGA | TGCTGTTGGCAAACCGTCTGT |

## ESC-to-EpiLC induction

A detailed protocol is described in [31]. After feeder depletion, 1.3 x $10^5$ ESCs were plated in the well of a 12-well plate pre-coated with fibronectin for 1 hr at 37°C. Medium was changed every day.

## Monolayer differentiation

After feeder depletion, 7 x $10^4$ ESCs were plated in the well of a 24-well plate pre-coated with gelatin. Cells were incubated with ESGRO Complete Basal medium (Millipore, Germany).

## Supporting information

**S1 Fig. DDX6 expression in early embryos.** (A) E6.5 embryo frozen section IHC for DDX6 and DCP1A (Scale: 50 μm for lower magnification, 10 μm for higher magnification). DAPI in blue. (B) E7.5 embryo frozen section IHC for BRACHYURY & DDX6 (Scale: 50 μm). (C) (1–3) E8.5 embryo frozen section IHC for DDX6. (4) Image of an E8.5 embryo expressing DDX6-mCherry (Scale: 100 μm for 1; 30 μm for 2–4). DAPI in blue. (D) Variation in the

morphology of E8.5 *Ddx6* KO embryos. (1–3) Lateral view of embryos that were younger than the head-fold stage. (4) An embryo with a markedly severe phenotype. It lacks the entire mesoderm. (5–6) Embryos with a head-fold structure and shortened primitive streak. (Scale: 200 μm). (E) Variability of E9.5 *Ddx6* KO embryos. (1) lateral view. (2–3) dorsal view. (Scale: 200 μm). (F) Whole-mount ISH of E8.5 embryos with a *Brachyury* probe (Scale: 100 μm). Representative images of the posterior and ventral view of the embryos, which were not shown in Fig 2A.
(PDF)

**S2 Fig. RNA-seq analyses of E8.5 *Ddx6*$^{\triangle/\triangle}$ embryos.** (A) Gene ontology (GO) term enrichment analysis of the most upregulated and downregulated genes. (B) RNA-seq data comparing expression of negative regulators of the BMP pathway in E8.5 *Ddx6* KOs with that in E8.5 WT. (C) Classification of types of the upregulated negative regulators of BMP signaling. (D-G, I-J) RNA-seq data comparing the expression of several key genes in E8.5 *Ddx6* KOs with that in E8.5 WT. (D) NSC and radial glial cell (NPC) markers. (E) Genes related to neuron-restricted intermediate progenitors and differentiated neurons. (F) Primitive streak and early mesoderm-related genes. (G) Differentiated mesoderm and endoderm markers. PSM: paraxial mesoderm, LPM: lateral plate mesoderm, DE: definitive endoderm. (I) Pluripotency marker genes. (J) BMP signaling components. (H) qRT-PCR analysis of some key genes in *Ddx6* KO E8.5 embryos. Most embryos used for this analysis were younger than the head-fold stage. Mean ± SEM. Significance was calculated by the Wilcoxon rank-sum nonparametric test (n = 10, 8, 12, 12, 8, 9, 10, 9, 8, 13, 8, 10, 5, and 7 in order of genes listed) (* at the $\alpha$ = 0.05 significance level, ** $\alpha$ = 0.01).
(PDF)

**S3 Fig. P-body expression in pluripotent cells and the effects of each gene KO on P-body formation.** (A) ICC of DDX6 and the P-body marker DCP1A during the ESC-to-EpiLC induction period (Scale: 20 μm). (B) Distinct granular P-bodies disappeared in the absence of DDX6. ICC of DCP1A, a P-body marker, in ESCs (Scale: 10 μm). (C) A scheme of genetic dissection of the DDX6-mediated RNA regulatory pathways. Three major DDX6-mediated pathways were disrupted by knocking out the key gene of each pathway. (D) P-bodies in ESCs were affected by the deletion of each gene. ICC of DDX6, a P-body marker, in ESCs (Scale: 10 μm).
(PDF)

**S4 Fig. Examination of Nodal signaling in *Ddx6*$^{\triangle/\triangle}$ ESCs and the comparison of SMAD2 and SMAD3 activity.** (A) Western blot of endogenous SMAD2/3 and phosphorylated SMAD2 in mESCs. Four conditions: SB (SB431542: TGF-β receptor ALK5 inhibitor)/—(non-treated)/ Activin 1-hr treatment/ Activin 2-hr treatment. The quantified signal intensity of the band is displayed in the right-side graph (n = 8). (B) Venn diagram showing the number of peaks bound by SMAD3 and SMAD2 from ChIP-seq performed for embryoid bodies [117]. (C) Binding score of MACS2 & STRING for SMAD3 and SMAD2 to genes of interest. Mean (± SEM).
(PDF)

**S5 Fig. The phenotypes of *Eif4enif1* KO and *Dcp2* KO.** (A) E9.5 *Eif4enif1* KO (Scale: 500 μm, n = 7) and *Dcp2* KO (Scale: 500 μm, n = 3) embryos with a littermate control. (B) Comparison of gene expression between *Ddx6* KO and *Eif4enif1* KO. qRT-PCR analysis of pluripotency genes during the EpiLC induction period. Each bar represents the relative expression of KO cells to WT cells at the indicated time point. Mean ± SEM. Student's t-test (n ≥ 3) (*p ≤ 0.05, **p ≤ 0.01, ***p ≤ 0.001, ****p ≤ 0.0001). (C) Integrative Genomics Viewer (IGV) snapshot of

each locus to confirm successful targeting.
(PDF)

**S1 Data. The numerical data.**
(XLSX)

**S1 Table. Gene sets that are highly enriched in three different conditions (Top11~20).**
(PDF)

**S2 Table. Gene sets that are differentially expressed only in *Ddx6* KO ESCs.**
(PDF)

**S3 Table. Gene sets that are differentially expressed only in *Dgcr8* KO ESCs.**
(PDF)

**S4 Table. Gene sets that are differentially expressed only in *Eif4enif1* KO ESCs.**
(PDF)

**S5 Table. Gene sets that are differentially expressed only in *Dcp2* KO ESCs.**
(PDF)

# Acknowledgments

We appreciate Prof. Ken Kurokawa and Associate Prof. Hiroshi Mori for supporting the transcriptomic analyses, Dr. Danelle Wright for proofreading, and Assistant Prof. Yuzuru Kato for fruitful discussion. We also thank Ms. Kumiko Inoue for supporting the mouse care and Ms. Yuko Sakakibara, Ms.Noriko Yamatani, and Mr. Makoto Kiso (Division for development of genetic-engineered mouse resource, NIG) for helping with experiments.

# Author Contributions

**Conceptualization:** Jessica Kim, Yumiko Saga.

**Data curation:** Masafumi Muraoka.

**Formal analysis:** Jessica Kim, Masafumi Muraoka, Hajime Okada.

**Investigation:** Jessica Kim, Atsushi Toyoda, Rieko Ajima.

**Methodology:** Jessica Kim, Rieko Ajima, Yumiko Saga.

**Resources:** Yumiko Saga.

**Supervision:** Rieko Ajima, Yumiko Saga.

**Validation:** Jessica Kim.

**Visualization:** Jessica Kim, Masafumi Muraoka, Hajime Okada.

**Writing – original draft:** Jessica Kim, Yumiko Saga.

**Writing – review & editing:** Jessica Kim, Rieko Ajima, Yumiko Saga.

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
