## [Decision Letter · Decision Letter 0]

13 Jan 2022

Dear Dr Kim,

Thank you very much for submitting your Research Article entitled 'The RNA helicase DDX6 controls early mouse embryogenesis by repressing aberrant inhibition of BMP signaling through miRNA-mediated gene silencing' to PLOS Genetics. Due to the holidays and the consequent challenge to identify available reviewers, we apologize for the delay in reaching an editorial decision.

Your article has been evaluated by the journal's senior editors and by three independent peer reviewers. While the study confirms a primary role for DDX6 as a mediator of miRNA driven gene repression, as reported by Freimer et al. (*Elife*, 2018), it challenges a primary role for DDX6 in P body biogenesis and provides evidence for a misinterpretation of the data reported by DiStefano et al. (*Cell Stem Cell*, 2019). The research conducted extends on previous work on the role of DDX6 in mammalian embryonic development by characterizing the developmental defects associated with DDX6 loss *in vivo*. It also uncovers both morphological and molecular phenotypes associated with DDX6 loss *in vivo*, suggestive for a major role of decreased BMP and increased Nodal signaling as the causes underlying the reported defects. As you will see, there is an overall consensus regarding the potential interest and significance of your work. However, the reviewers raised some concerns that revolve around the statistical analysis of the data shown; the presentation and evaluation of the results, as well as the quality, organization, and labeling of the figures; the choice of timepoint for bulk RNA sequencing; the rather superficial characterization of the DGCR8 embryos; and the confusing and lengthy elaboration of tangential points in the Discussion.

Specifically, Reviewer 1 raises serious concerns regarding the statistical analysis of the data presented, which is based on the use of only two samples of each genotype and thus is unsound, especially given the large variability between the samples. The Reviewer also worries about the choice of timepoint for bulk RNA sequencing - when the wildtype and mutant embryos are already strikingly different by E8.5 - which makes it awkward to interpret primary defects *versus* secondary defects due to the absence of entire portions of the embryo. The Reviewer does appreciate that conducting analyses at earlier timepoints -when the wildtype and mutant embryos have yet to show striking phenotypes - could be technically challenging. Thus, the authors should acknowledge these limitations, move all embryonic RNASeq analysis to supplemental data and use these data only to drive follow-up experiments, such as stainings and qRT-PCRs. Also, greater emphasis should be given to differential expression analysis of the RNASeq of the EpiLCs, as they do not suffer from the issue of secondary defects associated with the morphological defects, on a minimum of 3 samples for each genotype. Furthermore, the Reviewer requests that a deeper characterization of DGCR8 embryonic phenotypes be conducted, when comparing Ddx6 and DGCR8 KO embryos; otherwise, any statements of phenotypic similarities between the two mutants should be more guarded. Lastly, the Reviewer notes that the Discussion needs substantial streamlining.

Reviewer 2 recommends that an additional difference should be highlighted in the Discussion when comparing the current study to the Di Stefano et al. paper. The authors should discuss the potential difference between acute depletion of DDX6 via siRNA or CRISPRi *versus* longer-term knockout that was obtained in this study. The two different experimental approaches could be a potential source of disagreement – indeed, acute *versus* prolonged DDX6 depletion may largely differ. This Reviewer recommends the addition of a rescue experiment using a transgene that restores DDX6 function in knockout cells, which would provide a strong demonstration that the phenotypes reported in this work are specific to DDX6 loss. Without this additional demonstration, the Reviewer expresses some concerns with a possibility remaining that other factors might be mediating the difference between the conclusions reached in this study and prior work, due to different reagents and experimental systems used. 

Reviewer 3 recommends to strengthen the demonstration of an instructive role of BMP signaling in causing the reported Ddx6 KO phenotype by providing evidence also at the protein level, through IF staining of embryos for P-SMAD159. Given the most significant conclusion reached that diminished BMP and excess Nodal signaling are the underlying molecular defects causative of the phenotype, directly measuring those signaling pathways is of great importance, as concomitantly remarked also by Reviewer 1. Reviewer 3 also recommends some type of validation of successful KO of Eif4enif1 and Dcp2, which should be included.

Lastly, all three Reviewers raise concerns about the figures, regarding their quality, organization, plotting of the data, and/or labeling of the panels, which should be improved. For example, some of the images are of poor quality and should be improved; all embryo panels should make clear the developmental stage; figures showing composite images should separate the individual panels by white spaces.

The article and the critiques from the three reviewers have now been discussed among members of the editorial board with appropriate expertise. At present, we all agree that the work required to address the reviewers' (and our) concerns requires a ‘Major Revision’. If additional information is obtained in response to all of the critiques raised by the Reviewers, we will be willing to evaluate a revised manuscript that incorporates all of the revisions that have been requested. However, we caution that eventual success at PLOS Genetics will require satisfactorily addressing all of the main concerns that were raised by the Reviewers.

If you decide to revise the manuscript for further consideration at PLOS Genetics, please aim to resubmit within the next 60 days, unless it will take extra time to address the concerns of the reviewers, in which case we would appreciate an expected resubmission date by email to plosgenetics@plos.org.

[LINK]

We are sorry that we cannot be more positive about your manuscript at this stage. Please do not hesitate to contact us if you have any concerns or questions.

Yours sincerely,

Licia Selleri

Associate Editor

PLOS Genetics

Gregory Barsh

Editor-in-Chief

PLOS Genetics

Reviewer's Responses to Questions

**Comments to the Authors:**

Reviewer #1: The manuscript “The RNA helicase DDX6 controls early mouse embryogenesis by repressing aberrant inhibition of BMP signaling through miRNA-mediated gene silencing” extends on previous work studying the role of DDX6 in mammalian embryonic development. In particular, it confirms a primary role for DDX6 as a mediator miRNA driven gene repression (Freimer et al. 2018), while challenging a primary role for DDX6 in P body biogenesis (Di Stefano et al, 2019). It also extends the previous work by characterizing the development defects associated with DDX6 in vivo. In doing so, they uncover gross and molecular phenotypes suggestive for a major role decreased BMP and increased Nodal signal underlying the resulting morphological phenotypes. Finally, the authors attempt to make a direct link between DDX6 and increased expression of BMP inhibitors through the deregulation of presumably miRNA regulated transcription factors.

General thoughts:

1) The molecular similarity between DDX6 and DGCR8 knockout ES/EpiLCs cells is quite convincing and consistent with the previous work (Freimer et al., eLIFE. 2018). The in vivo analysis is also suggestive of similarities between the two mutants, but is very limited in terms of the characterization of the DGCR8 KO embryos.

2) Showing that alternative mutants of P body biogenesis have much less severe phenotypes does provide evidence for a misinterpretation of data by DiStefano et al (Cell Stem Cell, 2019). This is not surprising as it is known that P body disintegration can result through multiple mechanisms and is consequence rather than cause of the primary defect. For example, see https://www.ncbi.nlm.nih.gov/pmc/articles/PMC1900022/.

3) The most important contribution of this paper is the phenotypic characterization of DDX6 knockout embryos showing: a) severe, albeit variable, posterior defects, b) mesoderm differentiation defects including failed somitogenesis with associated bias toward endoderm markers, and c) premature neural differentiation. These phenotypes are suggestive of a major defect in BMP signaling with an associated expansion of Nodal signaling. These findings are consistent with previous work on the role of the predominant miRNAs in early development. That is, knockout of the miR-290/miR-302 clusters result in premature neural differentiation (Parchem et al. Cell Reports. 2015) and this family of miRNAs can directly suppress BMP inhibitors (Lipchina et al. G&D. 2011). Therefore, these data further the central role of DDX6 is in miRNA-mediated suppression, at least at this stage of development.

4) Overall, the paper provides some interesting insights that helps synthesize/clarify previous work, but there are a number of major issues that would need to be resolved.

Major issues:

1) Differential expression analysis using only two samples of each genotype is statistically unsound, especially given the huge variability between the two samples. Also, given that wt and mutant embryos are so strikingly different by E8.5, it is hard to interpret the bulk sequencing. That is, what is a primary defect vs. a secondary defect due to the absence of entire portions of the embryo. Capturing wt vs. mutant earlier when the wt and mutants have yet to show striking phenotypes would be much more interpretable. I appreciate that this could be technically difficult. At minimum, authors should acknowledge limitations and push all embryonic RNA seq analysis to supplement and simply use it to drive follow-up experiments such as stainings and qRT-PCRs. Greater emphasis could be given to DE analysis of the RNA-seq of the EpiLCs as they do not suffer from the issue of secondary defects associated with the morphological defects. However, again, there needs to be a minimum of 3 samples for each genotype to do such analysis.

2) Given the most significant conclusion the authors make is diminished BMP and excess Nodal signaling, they should directly measure those signaling pathways using relevant phosphoSMAD antibodies including IF staining of embryos and Westerns of EpiLCs.

3) Either a more thorough characterization of DGCR8 embryos should be performed or any statements of phenotypic similarity should be guarded. For example do DGCR8 embryos show similar defects in the domains of Nodal and EOMES, premature SOX1, somite morphogenesis, etc. Given that the similarity is a major conclusion of the paper, a more thorough characterization is preferred.

4) Figure panels are poorly organized and labeled (e.g. all embryo panels should make clear the developmental stage). Many of the images are rather low quality, although the phenotype is typically clear regardless. Still, it would be nice to get better images for publication. Also need to provide quantification such as how many embryos were analyzed and shows particular phenotype including morphological and staining defects.

5) Analysis in Fig. 7F and G is confusing and superficial. It should be removed or thoroughly followed up.

6) Given that the paper is essentially a DDX6 paper, the introduction should provide more background on DDX6 including previous biochemical work studying the molecular regulation of mRNA stability/ translational regulation by DDX6.

7) Conclusion is confusing and often tangential. Should focus on main points of paper: Points 1 through 3 under general comments above. For example the extensive discussion of DDX6 roles on the ISG pathway comes out of nowhere and seems irrelevant. Also, the discussion of Fig. 7F and G should be removed along with the data unless a much more thorough analysis (i.e. follow-up experiments) is done, which does not seem realistic.

Reviewer #2: Kim et al sought to determine the role of the RNA chaperone DDX6 in mouse development. They knocked out DDX6 and a variety of related genes and explored their impact on early mouse development in vivo and epiblast-like cells in vitro. The authors performed diverse assays to measure development in vitro and in vivo and connect gene expression changes with resulting phenotypes. The authors also provide a scholarly discussion on how their findings differ from prior work. The experiments are well executed in multiple systems, the authors are careful and rigorous in interpretation, and the findings are clearly discussed in this strong paper.

Major comments:

When comparing to Di Stefano et al, it seems that another difference is the acute depletion of DDX6 via siRNA or CRISPRi versus longer term knockouts in this work. This might be included in the discussion as another potential source of disagreement between the two studies – that acute versus prolonged DDX6 depletion may differ.

Adding a rescue experiment to Figure 5 where DDX6 is restored to knockout cells using a transgene would be a strong demonstration that the phenotypes in this work are specific to DDX6. Without this, the possibility remains that some unknown factor is mediating the difference between the conclusions in this work and prior work.

Minor comments:

Is Fig1a E7.5 panel a composite image of multiple images? If so, please separate the individual images by whitespace.

Reviewer #3: This manuscript by Kim et al demonstrates a critical requirement for RNA helicase DDX6 in early mouse development. The authors show that both constitutive and conditional deletion of Ddx6 gene results in gastrulation defects associated with inhibited BMP and increased Nodal signaling. Specifically, loss of DDX6 results in upregulation of BMP inhibitor and neuronal lineage marker transcription, which they validate using an in vitro model. Furthermore, the authors delete various epistatic interactors of DDX6 and could show that loss of DGCR8 most closely mimics DDX6 loss consistent with a role of DDX6 in miRNA-mediated translational repression underlying the developmental defects.

I think this manuscript is of importance to the field as it constitutes a first genetic analysis of DDX6 function in early mouse development and will thus be a useful contribution to our understanding of the framework of miRNA function in early development. I have no major issues with the main conclusions of the paper and only have minor comments (see below).

1. Line 167 “…genes of major developmental processes, especially the formation of mesoderm derivatives, were downregulated in Ddx6 KO (Fig. 1B)”: Not clear which GO categories are referred to. Examples would help.

2. As is, evidence for decreased BMP signaling is mainly consequential or via RNA. Protein evidence such as through staining for P-SMAD159 (see e.g. Senft et al 2019 Nature Communications, PMID: 30842446) would help strengthen the instructive role of BMP signaling in the observed Ddx6 KO phenotype.

3. Embryo stage should be included in ALL panels (e.g. Figure 3A, D, E and other figures)

4. Line 240: Reference to panels D,E missing.

5. Figure 5H: The way the data is plotted does could either suggest SMAD1/5 targets are de-repressed in Ddx6 KOs (as suggested by the authors) or alternatively that these genes are not repressed in Ddx6 KOs upon differentiation. A plot such as in 5D showing both WT and KO levels would clarify this.

6. RNA-seq (or other type of) validation of successful KO of Eif4enif1 and Dcp2 should be included.

7. Lines 456-458: This section of how putative BMP inhibitor regulating TFs were categorized would be useful in the respective results section and then discussed as is.

**Have all data underlying the figures and results presented in the manuscript been provided?**

Reviewer #1: **No: **As far as I can tell, none of the numerical data has been provided. GSE numbers are provided for RNA-seq, but is inaccessible without a token, which was not provided.

Reviewer #2: Yes

Reviewer #3: Yes

PLOS authors have the option to publish the peer review history of their article (what does this mean?). If published, this will include your full peer review and any attached files.

Reviewer #1: No

Reviewer #2: No

Reviewer #3: No

---

## [Decision Letter · Decision Letter 1]

22 Jun 2022

Dear Dr Kim,

Thank you very much for submitting a revised version of your manuscript entitled: "The RNA helicase DDX6 controls early mouse embryogenesis by repressing aberrant inhibition of BMP signaling through miRNA-mediated gene silencing" for review at PLOS Genetics. 

Your revised article has been evaluated by the journal's senior editors and by two independent peer reviewers. While your research confirms a primary role for DDX6 as a mediator of miRNA driven gene repression, as reported by Freimer et al. (*Elife*, 2018), it challenges a primary role for DDX6 in P body biogenesis and provides evidence for a misinterpretation of the data reported by DiStefano et al. (*Cell Stem Cell*, 2019). Hence your study will be of high interest to the field in relationship to the ongoing debate on the roles of P-bodies in post-transcriptional regulation, translational repression, and/or mRNA decay, which remain contentious.

In your revised paper all the major critiques that were raised by the reviewers have been addressed. The rigor and clarity of the manuscript have been enhanced by the addition of multiple experiments of high quality; the style and structure of the text have been improved; and the overall message has been further clarified. Lastly, the figures, their quality, organization, plotting of the data, and/or labeling of the panels, have been substantially improved. 

Specifically, upon the original review of the manuscript, Reviewer 1 worried about the choice of timepoint for bulk RNA sequencing - when the wildtype and mutant embryos are already strikingly different by E8.5 - which made it awkward to interpret primary defects versus secondary defects due to the absence of entire portions of the embryo. Thus, the Reviewer requested the authors acknowledge these limitations, move all embryonic RNASeq analysis to supplemental data and use these data only to drive follow-up experiments, such as stainings and qRT-PCRs. In response to this criticism, the RNA-seq data generated from E8.5 embryos were moved to the supplemental data and more detailed analyses on ESC-derived samples were conducted in the current manuscript. Also a deeper and more complete characterization of the phenotypes of Dgcr8 KO embryos was now conducted, substantially strengthening the overall conclusions of this part of the study.

Regarding the critiques raised by Reviewer 2, concerns about the potential differences between acute depletion of DDX6 via siRNA or CRISPRi versus longer-term knockout that was obtained in this study have now been addressed and any potential misunderstandings have been clarified. No claims are made that the phenotype described in the current manuscript is different from that described in the Di Stefano paper, but instead it is clearly stated that all differences reside in the interpretation of the dependency of DDX6 function on P-bodies. 

As for the suggestions of Reviewer 3, additional Western blot analyses were conducted to demonstrate that BMP signaling is indeed reduced in Ddx6-KO cells and additional validation data were provided for each gene KO.

Lastly, the quality of the figures and the presentation layout were strongly improved. Overall, all the revisions have significantly strengthened the current manuscript.

There are only a few minor points that should be further addressed before publication, as detailed in the Critiques to the Authors that follow below. In summary, Reviewer 2 requests that a few additional considerations be added to the Discussion of the revised manuscript. For example, the Reviewer suggests to include additional explanations that should consider other reasons why the roles of Dcp2 and Eif4enif1 differ from those of DDX6, beyond their role in P-body formation, e.g. they could have molecular functions of their own that are different than DDX6’s functions. The Reviewer incisively states that it is very hard to assign a specific function to a biomolecular condensate, as opposed to assigning a molecular function to the entities that form the biomolecular condensate. Indeed, DDX6, Eif4enif1, and Dcp2 are contained in P-bodies, but in addition they all have independent and distinct molecular functions. As a result, as Reviewer 2 remarks, differentiating molecular functions of the single proteins from functions of the biomolecular condensates is an extremely challenging task. These considerations should be included and considered in the Discussion, which should be further revised.

Lastly, Reviewer 3 recommends a few additional minor changes to some of the figures and plots, which should also be incorporated into the final version of the manuscript. 

The revised article and the critiques from the two reviewers have been discussed among members of the editorial board with appropriate expertise. At present, we all agree that a few additional ‘Minor Revisions’ are required in response to the critiques currently raised by the Reviewers. These are editorial revisions that will not require additional experiments. We will be happy to evaluate a revised manuscript that incorporates these additional editorial revisions. 

[LINK]

Yours sincerely,

Licia Selleri

Associate Editor

PLOS Genetics

Gregory Barsh

Editor-in-Chief

PLOS Genetics

Reviewer's Responses to Questions

**Comments to the Authors:**

Reviewer #2: Kim et al prepared a good reply to review comments, including various clarifications and experiments to address reviewer comments. There remain some uncertainties as to why this and prior work differ, but the experiments and analyses are robust and in my opinion this work should be published. My one comment that would improve the paper in its current version is to add a discussion of other reasons why Dcp2 and Eif4enif1 might differ from DDX6 beyond just their role in P-body formation, for example because they have molecular functions of their own that are different than DDX6’s. It is very hard to assign a function specifically to a biomolecular condensate as opposed to a molecular function of the entities that form a biomolecular condensate. DDX6, Eif4enif1, and Dcp2 are in P-bodies and also all have independent and distinct molecular functions, so differentiating molecular functions from functions of biomolecular condensates requires more than knockouts. This could be included in the discussion.

Reviewer #3: My comments were addressed satisfactorily.

- Fig. 5H would be more legible if split into 4 plots (one per mRNA)

- “brackets” used for indicating significance on various plots (e.g. Fig. 4H) are stylistically distracting/unnecessary, asterisks above the compared timepoints would suffice

**Have all data underlying the figures and results presented in the manuscript been provided?**

Reviewer #2: Yes

Reviewer #3: Yes

PLOS authors have the option to publish the peer review history of their article (what does this mean?). If published, this will include your full peer review and any attached files.

Reviewer #2: No

Reviewer #3: No

---

## [Decision Letter · Decision Letter 2]

11 Aug 2022

Dear Dr Saga,

We are pleased to inform you that your manuscript entitled "The RNA helicase DDX6 controls early mouse embryogenesis by repressing aberrant inhibition of BMP signaling through miRNA-mediated gene silencing" has been editorially accepted for publication in PLOS Genetics. Congratulations!

Yours sincerely,

Licia Selleri

Academic Editor

PLOS Genetics

Gregory Barsh

Editor-in-Chief

PLOS Genetics

Comments from the reviewers (if applicable):

Reviewer's Responses to Questions

**Comments to the Authors:**

Reviewer #2: Thanks to the authors for their work in addressing all reviewer comments - this work should be published.

**Have all data underlying the figures and results presented in the manuscript been provided?**

Reviewer #2: Yes

PLOS authors have the option to publish the peer review history of their article (what does this mean?). If published, this will include your full peer review and any attached files.

Reviewer #2: No

**Data Deposition**

http://datadryad.org/submit?journalID=pgenetics&manu=PGENETICS-D-21-01565R2

**Press Queries**

---

## [Editor Report · Acceptance letter]

21 Sep 2022

PGENETICS-D-21-01565R2 

The RNA helicase DDX6 controls early mouse embryogenesis by repressing aberrant inhibition of BMP signaling through miRNA-mediated gene silencing 

Dear Dr Saga, 

We are pleased to inform you that your manuscript entitled "The RNA helicase DDX6 controls early mouse embryogenesis by repressing aberrant inhibition of BMP signaling through miRNA-mediated gene silencing" has been formally accepted for publication in PLOS Genetics! Your manuscript is now with our production department and you will be notified of the publication date in due course.

With kind regards,

Agnes Pap

PLOS Genetics

On behalf of:
